# Planarian EGF repeat-containing genes *megf6* and *hemicentin* are required to restrict the stem cell compartment

Nicole Lindsay-Mosher[1,2], Andy Chan[1,2,3], Bret J. Pearson[1,2,4]*

**1** The Hospital for Sick Children, Program in Developmental and Stem Cell Biology, Toronto, Ontario, Canada, **2** University of Toronto, Department of Molecular Genetics, Toronto, Ontario, Canada, **3** School of Biomedical Sciences, LKS Faculty of Medicine, Pokfulam, Hong Kong SAR, China, **4** Ontario Institute for Cancer Research, Toronto, Ontario, Canada

* bret.pearson@sickkids.ca

**Data Availability Statement:** All relevant data are within the manuscript and its Supporting Information files.

## Abstract

The extracellular matrix (ECM) is important for maintaining the boundaries between tissues. This role is particularly critical in the stem cell niche, as pre-neoplastic or cancerous stem cells must pass these boundaries in order to invade into the surrounding tissue. Here, we examine the role of the ECM as a regulator of the stem cell compartment in the planarian *Schmidtea mediterranea*, a highly regenerative, long-lived organism with a large population of adult stem cells. We identify two EGF repeat-containing genes, *megf6* and *hemicentin*, with identical knockdown phenotypes. We find that *megf6* and *hemicentin* are needed to maintain the structure of the basal lamina, and in the absence of either gene, pluripotent stem cells migrate ectopically outside of their compartment and hyper-proliferate, causing lesions in the body wall muscle. These muscle lesions and ectopic stem cells are also associated with ectopic gut branches, which protrude from the normal gut towards the dorsal side of the animal. Interestingly, both *megf6* and *hemicentin* knockdown worms are capable of regenerating tissue free of both muscle lesions and ectopic cells, indicating that these genes are dispensable for regeneration. These results provide insight into the role of planarian ECM in restricting the stem cell compartment, and suggest that signals within the compartment may act to suppress stem cell hyperproliferation.

## Author summary

The freshwater planarian maintains a large population of adult stem cells throughout its long lifespan. Although these stem cells are constantly dividing, the rate of division is tightly controlled to such a degree that planarians almost never develop neoplastic growths. In addition, the stem cells are located in a specific spatial compartment within the animal, although no known physical boundary keeps them in place. What mechanisms do planarians use to control the number, rate of division, and location of their stem cells? Here, we find that two EGF repeat-containing genes, *megf6* and *hemicentin*, are required to keep stem cells within their compartment. Although these two genes are

**Funding:** BJP was supported by Ontario Institute for Cancer Research (oicr.on.ca) Investigator grant #IA-026. NLM was supported by a student Restracomp award from the Hospital for Sick Children (www.sickkids.ca), as well as Canada Institutes for Health Research grant #PJT-159611 (www.cihr-irsc.gc.ca). AC was supported by the University of Toronto and Hong Kong University joint PhD exchange programme. The funders had no role in study design, data collection and analysis, decision to publish, or preparation of the manuscript.

**Competing interests:** The authors have declared that no competing interests exist.

expressed in different cell populations, we find that both are required to maintain the epithelial basal lamina. In the absence of either gene, stem cells can escape their compartment and migrate towards the skin of the animal, where they divide at an accelerated rate and cause lesions in the muscle. These results show that the extracellular matrix plays a role in limiting the boundaries of the stem cell compartment.

## Introduction

Long-lived animals require mechanisms to promote continued adult stem cell (ASC) self-renewal and potential while also suppressing neoplastic behaviour [1,2]. One mechanism to balance proliferation and tumor suppression is through the stem cell niche: the physical region capable of maintaining stem cell potential and proliferative capacity. The stem cell niche is comprised of two main factors: 1) differentiated cells, often important for producing signals such as growth factors; and 2) the extracellular matrix (ECM), the network of insoluble proteins that provides structure to the intercellular space [3–5]. A growing body of evidence supports the ECM as an important regulator of stem cell behaviors, including division, quiescence, and differentiation [3,6–8]. At the tissue level, the ECM also provides a physical barrier that prevents neoplastic stem cells from leaving their compartment and invading into other tissues, thereby slowing tumour progression and metastasis [4,9,10]. At the multi-tissue level, the ECM promotes regeneration and wound closure, particularly in regenerative species such as zebrafish and mouse strains with enhanced healing [11,12]. How the ECM carries out these diverse functions has yet to be fully understood.

The freshwater planarian, *Schmidtea mediterranea*, is a well-established model of adult stem cells and provides a powerful system in which to study the stem cell niche, *in vivo*. The planarian maintains a large population of ASCs called neoblasts, which are collectively pluripotent, serve to constantly replenish all tissue types, and are capable of renewing indefinitely [13–15]. Neoblasts are crucial to the planarian's regenerative abilities, which allow them to regrow all organs and tissues following amputation [16–18]. Despite the high rate of cell division of planarian ASCs, wild-type planarians do not develop tumours, and pathogenic hyperproliferation is rarely observed [19]. Neoblasts are located in the central parenchyma of the animal and stay in their own compartment, even though no known physical boundary keeps them in place [20]. When a single neoblast is transplanted into an irradiated animal with no stem cells, it will symmetrically expand to refill the parenchyma of the animal and stop doing so when the expanding population hits the extent of the niche [14]. Taken together, the strict spatial limits of neoblasts suggests the presence of robust regulatory mechanisms to restrict the division and migration of planarian ASCs, perhaps by a physical niche.

Little is known about what factors comprise the physical stem cell niche in planarians; however, several recent studies have demonstrated a role of the ECM in regulating the localization and behaviour of planarian neoblasts, hinting that the ECM may be a major component of the stem cell niche. Loss of *β1-integrin*, a transmembrane receptor that allows cells to interact with the ECM, results in mis-localization of stem cells during regeneration [21]. Both *β1-integrin* and the matrix metalloprotease *mmpa* are required for neoblast migration via the epithelial-to-mesenchymal transition (EMT) following injury [22]. The ECM may regulate localization of neoblasts by physically restricting their migration or by coordinating with growth factor signaling, as has been demonstrated in other systems [23]. Interestingly, epidermal growth factor receptor (EGFR) signaling has been shown to restrict neoblast proliferation, localization, and differentiation in planarians [24,25]. Yet, how the ECM may regulate neoblast behaviour, through EGFR signaling or otherwise, remains to be elucidated.

In a planarian RNAi screen of putative-ECM components, we identified two EGF repeat-containing genes required to restrict stem cells to within their compartment. One gene was found to encode *hemicentin*, a glycoprotein previously reported to maintain the boundaries between tissue compartments in planarians, while the other encodes *megf6*, a gene with very little functional characterization in any organism. Here, we find that *megf6* is an ancient gene, conserved among all animal superphyla, and is closely related to the well-characterized Draper/Ced-1/MEGF10 family of phagocytic receptors. However, *megf6* does not share the phagocytic role of the Draper/Ced-1/MEGF10 proteins, and is instead required to maintain the integrity of the basement membrane in planarians. We demonstrate that in the absence of either *megf6* or *hemicentin*, stem cells hyper-proliferate ectopically and cause lesions in the body wall muscle. Using transmission electron microscopy, we find that the epithelial basal lamina and muscle fiber position are disrupted following knockdown. Together, these results suggest that the basement membrane component of the niche may play a restrictive rather than a permissive role for stem cell proliferation. Interestingly, although knockdown of either gene has severe effects in homeostatic tissues, we find that both genes are dispensable for regeneration. These findings demonstrate the importance of the ECM in maintaining the boundaries of the stem cell compartment during homeostasis, and suggest that the stem cell niche plays a role in limiting proliferation.

## Results

### An extracellular matrix screen reveals genes required for basement membrane integrity

We performed an RNAi screen of putative-ECM components in planarians to determine how the ECM regulates neoblast function. Candidate genes were chosen using a homology-based approach, including genes with conserved domains known to be involved in ECM function in other organisms. We noted that two genes, both containing multiple EGF-repeat domains, produced identical, unusual morphological phenotypes upon knockdown: wrinkling of the epidermis as well as unpigmented spots on the dorsal side (Fig 1A). These two genes were identified as *megf6* and *hemicentin*; these findings confirmed recent observations that knock-down of *hemicentin* in planarians results in wrinkled epidermis [26]. The wrinkling phenotype was 100% penetrant after 9 feeds of RNAi (0/150 animals in control RNAi, 150/150 in *megf6* RNAi, 150/150 in *hemicentin* RNAi), whereas the appearance of unpigmented spots was only about 50–60% penetrant (0/150 animals in control RNAi, 86/150 in *megf6* RNAi, 72/150 in *hemicentin* RNAi). Knockdown planarians remained alive and exhibited normal locomotive and feeding behaviours up to 10 weeks after the first RNAi feed, indicating that these pheno-types were not lethal.

To further characterize *megf6* and *hemicentin*, we assayed the expression of these genes using whole-mount *in situ* hybridization (WISH). Expression of planarian *megf6* was strongest in the pharynx, though low-level expression was also detected in the epidermis (Fig 1B and 1C) [27–29]. This finding was corroborated by previously-published single-cell RNA-sequencing data (scRNAseq), which detected *megf6* expression in pharynx cells, but also detected low-level expression in epidermal and intestinal tissue (S1 Fig) [30]. By contrast, *hemicentin* expression was almost entirely in muscle cells (S1 Fig) [26]. Interestingly, within the pharynx, *hemicentin* was co-expressed with the muscle cell marker *collagen* whereas *megf6* expression was not, indicating that these two genes were expressed by different cell populations in the same location (Fig 1B and 1C, S1 Fig). WISH staining showed no decrease in mRNA expression of *hemicentin* following *megf6* knockdown and vice versa, ruling out the hypothesis that one gene

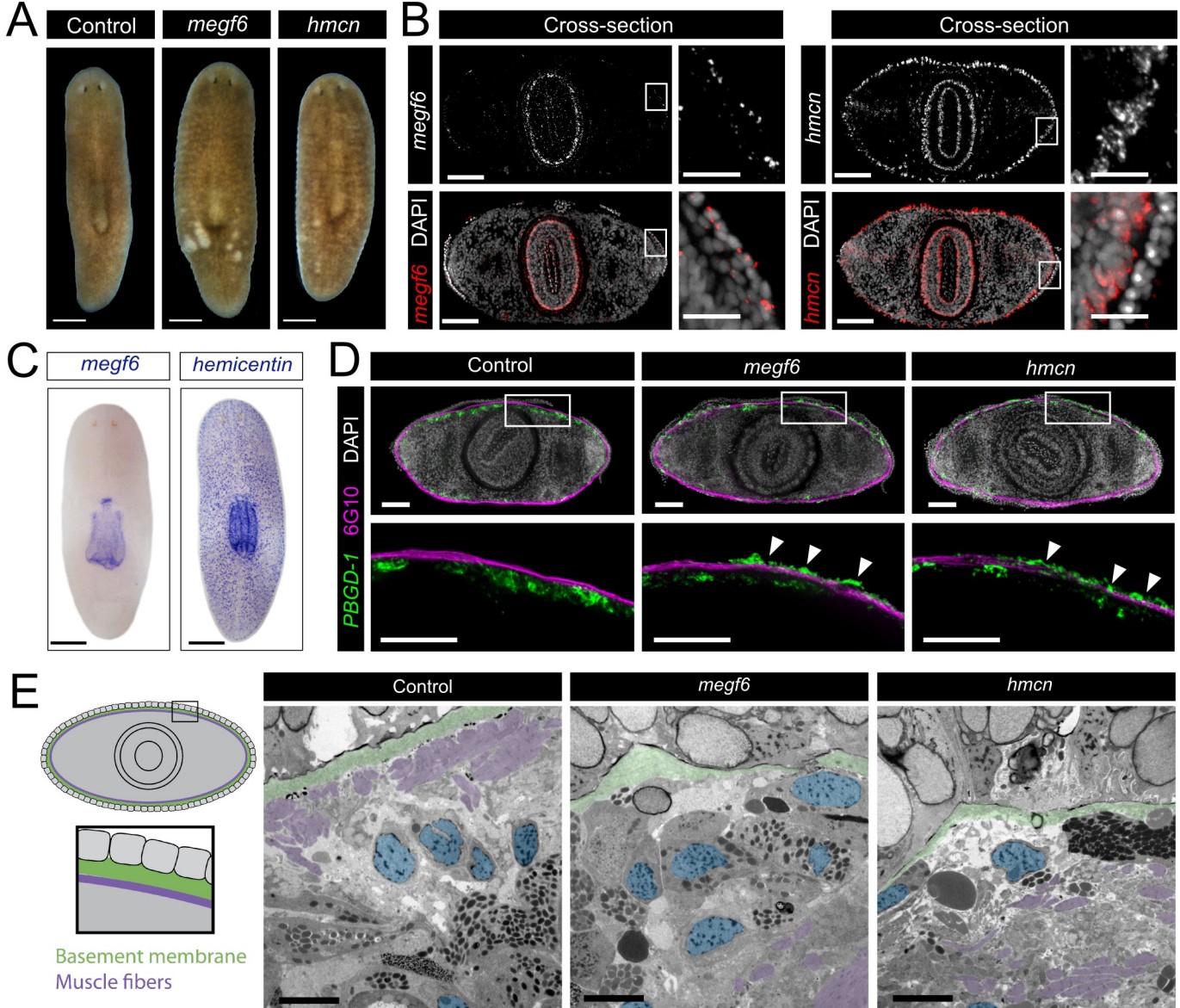

**Fig 1. Expression and knockdown phenotypes of *megf6* and *hemicentin*. A)** Live images of control, *megf6* and *hemicentin* knockdown planarians, dorsal side up (n = 150). **B)** Fluorescent *in situ* hybridization (FISH) of *megf6* and *hemicentin*, shown in cross-section through the pharynx. White boxes denote the magnified region shown on the right. **C)** Colorimetric RNA *in situ* hybridization of *megf6* and *hemicentin* in wild-type planarians. **D)** Transverse cross-sections of planarians with FISH for *PBGD-1*, a marker for immature pigment cells, and immunostained with 6G10 antibody, which marks muscle fibers (n ≥ 3). White boxes denote the magnified region shown below. Ectopic pigment cells are shown with white arrows. **E)** Transmission electron microscopy of the subepidermal region of control and knockdown planarians (n = 3). A diagram showing the region of the worm from which the images were taken is shown on the left. The basement membrane is coloured green, subepidermal nuclei are blue, and muscle fibers are purple. Scale bars: 250 μm in A, C; 100 μm in B, 25 μm in magnified regions; 100 μm in D; 5 μm in E.

simply regulates the expression of the other (S1 Fig). Given these results, we hypothesized that *megf6* and *hemicentin* both play a role in ECM integrity in planarians.

To examine the effects of *megf6* or *hemicentin* knockdown, and given an apparent increase in pigmentation in the epidermal wrinkles, we examined the localization of pigment cells. To determine the location of the pigment cells in relation to the body wall muscle layer, we visualized muscle fibers using the monoclonal antibody 6G10, which labels circular and diagonal

fibers, and pigment cells using the marker gene *PBGD-1* [31–33]. Transverse cross-sections revealed that in control animals pigment cells were located basal to the muscle layer, whereas knockdown animals displayed a second layer of ectopic pigment cells between the muscle and the epidermis (Fig 1D). We hypothesized that the wrinkles observed in the epidermis of *megf6* and *hemicentin* knockdown animals were caused by ectopic cells located between the muscle layer and the epidermis, causing the epidermis to become deformed. By transmission electron microscopy (TEM), ectopic nuclei were observed between the basement membrane and the muscle fibers of knockdown animals, recapitulating the results seen by WISH (Fig 1E). The muscle fibers of knockdown worms were located several cell diameters from the basement membrane, whereas in controls the basement membrane directly contacted the muscle fibers (Fig 1E). Additionally, the basement membrane was clearly disorganized in knockdown worms; either much thinner, less dense, or completely absent in different areas (Fig 1E). From this, we concluded that *megf6* and *hemicentin* are required for the maintenance of the basement membrane of the epidermis, and that in the absence of either gene the attachment of both the epidermis and the body wall muscle to the basement membrane is disrupted.

## *megf6* is an uncharacterized gene conserved in metazoans

The gene *hemicentin* has previously been identified as an ECM component in planarians, and has clear orthologs in *C. elegans*, *Drosophila*, and vertebrates. However, *megf6* has very little functional characterization in any organism. To determine the conservation of this gene, we performed reciprocal BLAST searches to vertebrates and found two possible orthologs: MEGF6, and the well-characterized phagocytic receptor MEGF10/11. To resolve the evolutionary history of these genes, we performed a Bayesian phylogenetic analysis (S2 Fig, Supplemental File 1). We found clear orthologs of both MEGF10/11 and MEGF6 in all superphyla, leading us to conclude that these two genes existed before the bilaterian split. Surprisingly, we found evidence of two separate, clade-specific gene loss events. First, we could not find MEGF6 in any insects, yet it is clearly present in crustaceans (shrimp) and chelicerates (scorpion); second, we found no MEGF10/11 orthologs in triclads (planarians), polyclads (marine flatworms), or *Macrostomum lignano*, a basal flatworm. These data support the loss of MEGF10/11 in the Platyhelminthes; therefore, we refer to the planarian gene identified in our screen as *megf6*.

Draper/Ced-1/MEGF10/11 plays an important role in cell corpse removal in *Drosophila*, *C. elegans*, and vertebrates [29,34,35]. Therefore, we asked whether *megf6* takes on this role in planarians, which lack a MEGF10/11 ortholog. Interestingly, planarians do have orthologs of many of the regulators of phagocytosis downstream of Draper/Ced-1/MEGF10/11 in flies and *C. elegans*, including *ABCA1*, *shark*, *ELMO*, and *gulp* [36–39]. However, upon RNAi knockdown we found that none of these genes reproduced the knockdown phenotypes of *megf6* (S3 Fig). These data suggest that the role of *megf6* is likely distinct from the closely related Draper/Ced-1/MEGF10/11 gene family.

## Multiple ectopic cell types are observed following knockdown of *megf6* or *hemicentin*

Pigment cells normally reside in the sub-epidermis basal to the muscle layer, and therefore would likely be directly affected by changes to the basement membrane of the epidermis. We asked whether cell types far from the basement membrane were also affected by *megf6* or *hemicentin* knockdown by staining for multiple cell type-specific gene markers. We found that cells appeared in ectopic locations regardless of the proximity of their normal location to the basement membrane. As previously observed following knockdown of *hemicentin* [26], *piwi-1*⁺

neoblasts were detected in clusters near the dorsal epidermis of *megf6* or *hemicentin* knock-down animals, whereas in control animals neoblasts were restricted to the parenchyma (Fig 2A). Cells stained with *prog-2* or *agat-3*, marking early and late progenitors of the epidermal lineage, respectively, were found adjacent to the epidermis independent of *piwi-1*$^+$ neoblasts, indicating that these cells were not merely the progeny of ectopic stem cells but also migrated ectopically (Fig 2B). In control animals, the normal organization of the epidermal progenitor cells recapitulated their lineage progression, with *piwi-1*$^+$ cells located in the deepest paren-chyma followed by their slightly more apical *prog-2*$^+$ progeny, and finally *agat-3*$^+$ cells closest to the epidermis. This organization appeared disrupted following knockdown of *megf6* or *hemicentin*, with *prog-2*$^+$ and *agat-3*$^+$ cells intermingled rather than ordered by cell type, indi-cating that localization of these cell types in the parenchyma was altered (Fig 2B). We also noted that ectopic *prog-2*$^+$ and *agat-3*$^+$ cells were sometimes located near ectopic neoblasts as previously observed, indicating that entire ectopic epithelial lineages may be produced apical to the muscle layer (Fig 2C) [26].

In addition to the epidermal lineage, we observed multiple ectopic neural cell types. Staining for *ovo* revealed clusters of ectopic photoreceptors located anteriorly and dorsally to the normal eyespots (Fig 2D). Interestingly, these cells always formed a cluster, and in some cases even appeared to form a structure reminiscent of the eyespot; thus, these cells retained some of their ability to self-organize despite their ectopic location, similar to when eye-positional information is altered in the eye field [40,41]. Staining of axonal tracts with the monoclonal antibody 1H6 showed that neurons in *megf6* or *hemicentin* knockdown ani-mals were able to form a grossly normal peripheral neural network (Fig 2E). However, cho-linergic neurons marked with *chat* were observed in the sub-epidermis outside of this network, indicating that ectopic neurons were also present following *megf6* or *hemicentin* knockdown (Fig 2E) [42].

## Ectopic neoblasts hyper-proliferate and differentiate

Unlike most of the ectopic cell types, ectopic *piwi-1*$^+$ cells in *megf6* or *hemicentin* RNAi ani-mals formed clusters. One possible explanation for this was that individual ectopic neoblasts proliferated and formed colonies outside of the normal neoblast compartment, a hypothesis supported by the previous observation of ectopic dividing cells in *hemicentin* knockdown animals [26]. To test whether ectopic *piwi-1*$^+$ cells were proliferative, we co-stained planari-ans with *piwi-1* and the G2/M-phase marker phosphorylated-histone H3 on serine 10 (H3P). With transverse cross sections taken at the same axial level, we were able to make a clear distinction between neoblasts located within the normal neoblast compartment and those in ectopic locations, and showed that both populations expressed H3P (Fig 3A). We quantified the proportion of *piwi-1*$^+$ cells co-labelled with H3P, and found that a higher pro-portion of ectopic neoblasts expressed H3P compared to their counterparts within the nor-mal compartment (Fig 3B). Ectopic neoblast clusters were also labelled with the thymidine analog BrdU after a 24-hour chase period, indicating that the cells had gone through S-phase during this time and not simply arrested at the G2/M transition (Fig 3C). Interest-ingly, whole-animal H3P staining as well as fluorescence activated cell sorting (FACS) anal-ysis demonstrated that on the level of the whole animal, proliferation was not overtly affected by *megf6* or *hemicentin* RNAi (S4 Fig). Together, these results suggested that ectopic neoblasts were hyperproliferating compared to neoblasts within the normal compartment.

Neoblasts in the normal stem cell niche display heterogeneity, such that cells throughout the *piwi-1*$^+$ population co-express markers of different lineages or known neoblast subtypes

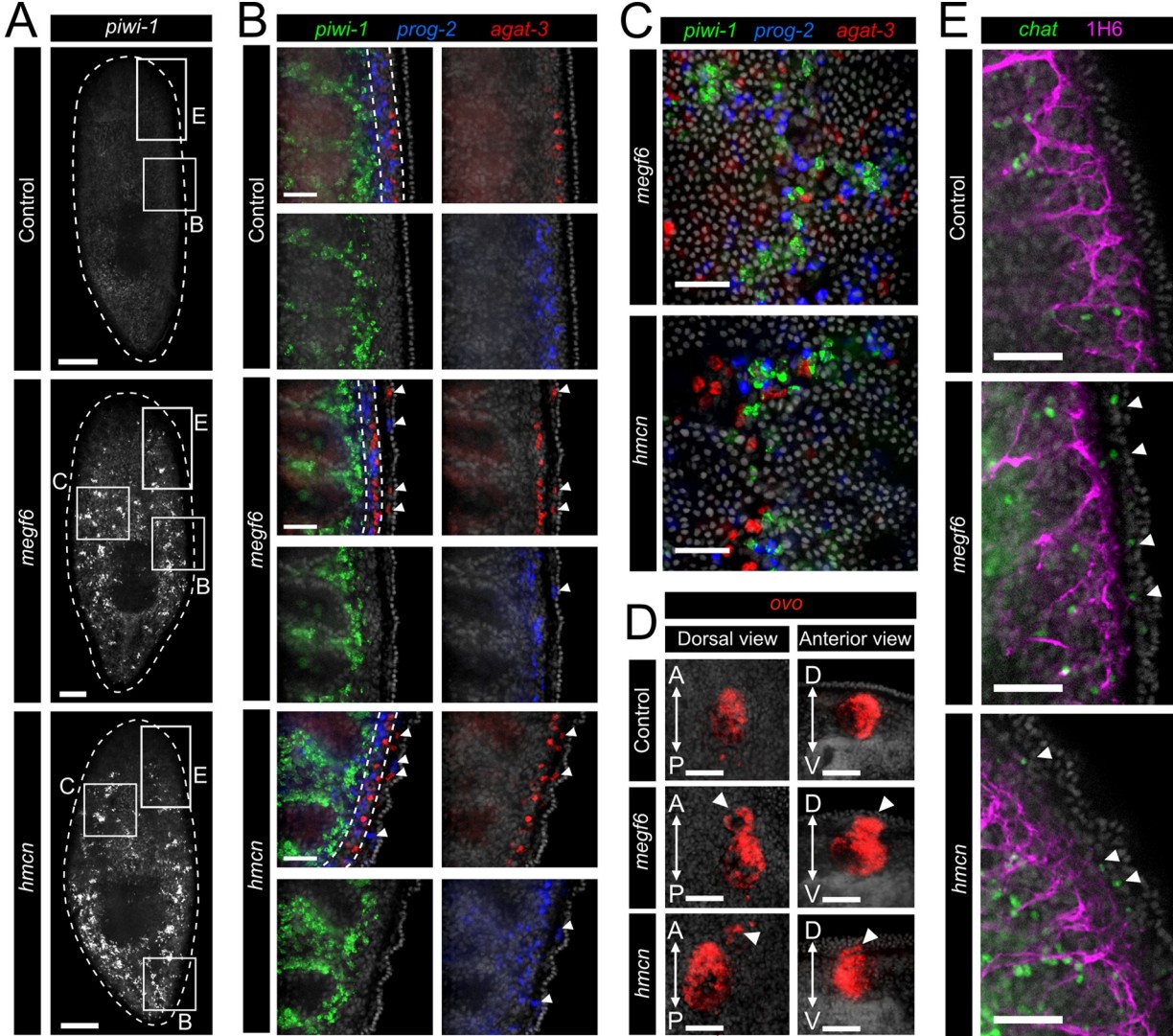

**Fig 2. Ectopic cell types in *megf6* or *hemicentin* knockdown. A)** Single confocal planes at the dorsal sub-epidermis of control, *megf6*, or *hemicentin* knockdown planarians stained with FISH for *piwi-1* (n ≥ 6). White boxes denote the regions used for panels B, C and E. **B)** Triple FISH for *piwi-1*, *agat-3*, and *prog-2* imaged in single confocal planes at the lateral edge of control, *megf6*, or *hemicentin* knockdown planarians (n ≥ 6). Ectopic *prog-2*+ and *agat-3*+ cells are shown with white arrowheads. The parenchymal region in which *prog-2*+ cells normally progress to *agat-3*+ cells is denoted with white dotted lines in the merged image. **C)** Single confocal planes showing *piwi-1*, *agat-3*, and *prog-2* at the dorsal epidermis in knockdown planarians (n ≥ 6). **D)** FISH for *ovo* with ectopic photoreceptors marked by white arrowheads (n = 4). Dorsal view shown on the right; anterior view (from transverse cross-sections) of the same eye shown on the left. A is anterior, P is posterior, D is dorsal, and V is ventral. **E)** FISH for *chat* co-stained with 1H6, an antibody marking peripheral axon tracts (n ≥ 7). Single confocal planes at the lateral edge of control, *megf6*, and *hemicentin* knockdown planarians are shown. Ectopic *chat*+ cells are marked with white arrowheads. Scale bars: 250 μm in A; 50 μm in B-E.

[43]. To determine whether a similar heterogeneity exists in the ectopic neoblast clusters, we stained for three marker genes for known neoblast subclasses. We found expression of the epidermal lineage marker *zfp-1*, the endodermal marker *hnf4*, and the putative marker of pluripotent neoblasts *tgs-1* in ectopic *piwi-1*+ clusters (Fig 3D) [15,43]. This indicated that multiple classes of neoblasts were present in the ectopic *piwi-1*+ population, supporting our hypothesis that these cells were capable of producing entire ectopic cell lineages. Quantification of each neoblast subclass in the ectopic cells showed a significant decrease in the proportion of *tgs-1*+

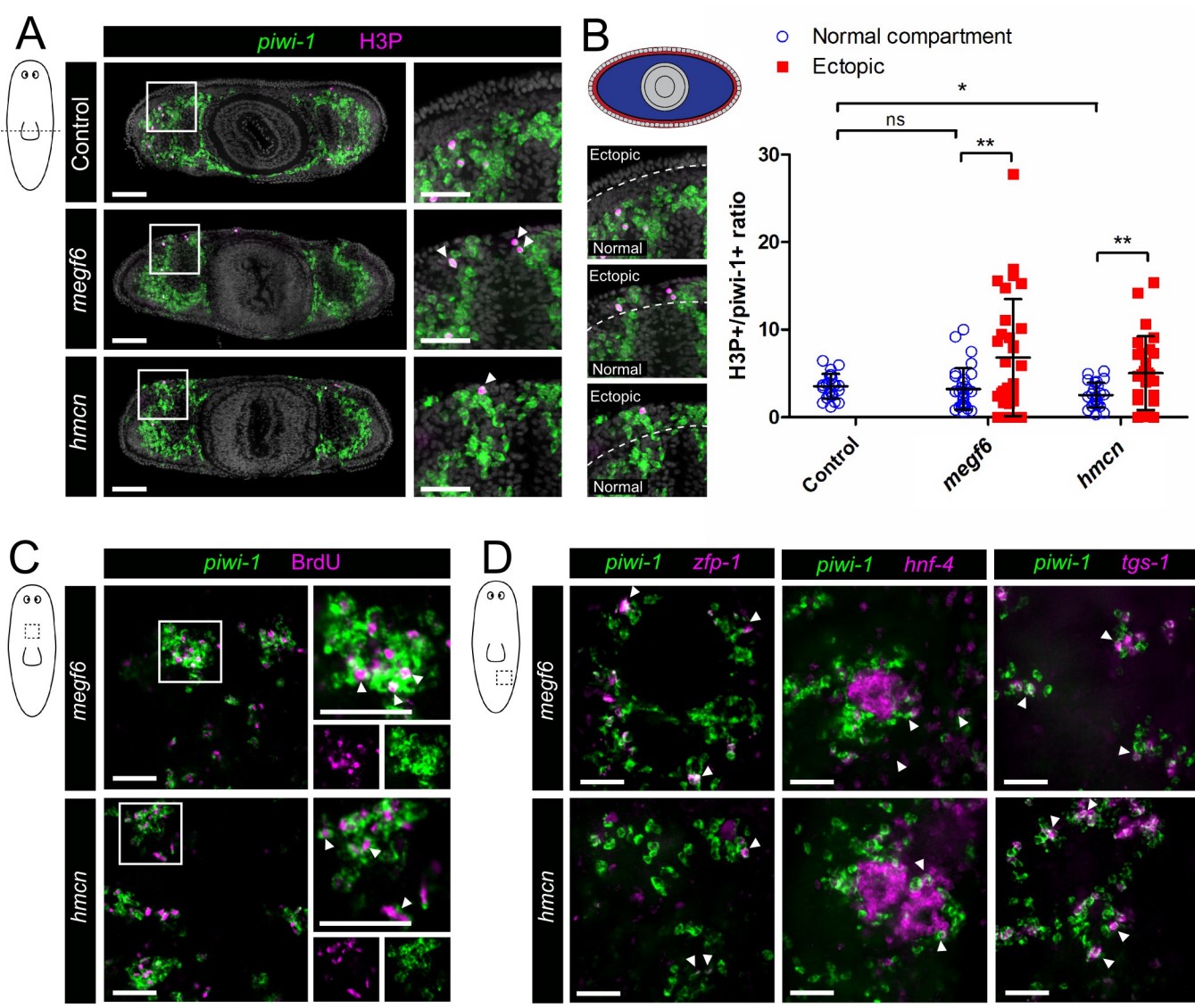

**Fig 3. Proliferation and lineage specification of ectopic neoblasts. A)** Transverse cross sections of control, *megf6*, and *hemicentin* knockdown planarians stained with FISH for *piwi-1* and the G2/M-phase marker phosphorylated-histone H3 on serine 10 (H3P). White boxes denote the magnified region, shown on the right. Ectopic *piwi-1*[+] cells expressing H3P are marked with white arrowheads. **B)** Percentage of *piwi-1*[+] cells expressing H3P in each condition, quantified in transverse cross-section and separated by location (n ≥ 24). Diagrams indicating the 'ectopic' and 'normal' compartments are shown on the left. White dotted lines denote the boundary between the two compartments. **C)** FISH for *piwi-1* combined with BrdU after a 24-hour chase in knockdown planarians (n ≥ 5). Single confocal planes at the level of the dorsal sub-epidermis show clusters of ectopic *piwi-1*[+] cells. White boxes show the magnified region on the right, and white arrowheads denote ectopic *piwi-1*[+] labelled with BrdU. **D)** Double FISH for *piwi-1* combined with *zfp-1*, *hnf4*, or *tgs-1* (n ≥ 8). Ectopic *piwi-1*[+] cells expressing *zfp-1*, *hnf4*, or *tgs-1* are marked with white arrowheads. Scale bars: 100 μm in A, 50 μm in magnified regions; 50 μm in C, D. Error bars are standard deviations. *p < 0.05, **p < 0.01, ns = not significant (Welch's t-test).

cells compared to the normal compartment, potentially indicating that more of these cells were lineage-committed (S5 Fig). We also observed a trend towards fewer *zfp-1*[+] cells and more *hnf4*[+] cells in the ectopic clusters, suggesting that these cells may have been biased towards the endodermal lineage (S5 Fig). In line with this, several of the ectopic *piwi-1*[+] clusters were centered around a group of *hnf4*-expressing cells, marking ectopic differentiated gut tissue (Fig 3D).

## Ectopic stem cells correlate with muscle lesions and ectopic gut branches

Although most of the ectopic cell types we observed in *megf6* or *hemicentin* knockdown animals were independent of the ectopic neoblast clusters, we did notice two major tissue defects correlated with ectopic *piwi-1*+ cells. Using the antibody 6G10 to visualize muscle fibers, we observed severe lesions in the body wall muscle, which in all cases corresponded to the location of ectopic *piwi-1*+ cells (Fig 4A) [31]. We also observed the presence of ectopic gut branches, marked by *mat*, which were always surrounded by *piwi-1*+ cells and located within a large muscle lesion (Fig 4A and 4B). The presence of ectopic *piwi-1*+ clusters was accompanied by lesions in the muscle and vice versa, whereas only the largest lesions contained an ectopic gut branch. Ectopic *piwi-1*+ cell clusters and muscle lesions were observed in all knockdown animals while ectopic gut branches were observed in 60–90% of *megf6* or *hemicentin* RNAi worms (0/10 animals in control RNAi, 9/10 in *megf6* RNAi, 6/10 in *hemicentin* RNAi). Although ectopic neoblasts and muscle lesions appeared on both the dorsal and ventral sides, the number and size of the lesions were greater on the dorsal side, and gut branches were only observed protruding dorsally (Fig 4C, S6 Fig). We also noted that lesions were mostly absent from the head region as well as the area dorsal to the pharynx (Fig 4A).

Despite the severe defects in body wall muscle, the density of *collagen*+ body wall muscle cells increased slightly in *megf6* and *hemicentin* RNAi worms compared to controls, which could indicate proper homeostatic production of muscle cells or no increase in muscle cell death (S7 Fig). In line with this, TUNEL staining to mark apoptotic cells was not significantly increased in knockdown planarians (S7 Fig). Interestingly, although the body wall muscle was always absent where an ectopic gut branch was present, the ectopic gut tissue itself was surrounded by enteric muscle, similar to the normal gut (Fig 4B). These observations suggested that the production of muscle cells and formation of the muscle fiber network could both still occur in *megf6* and *hemicentin* knockdown animals. The slight increase in muscle cell density could be due to increased muscle cell production in an effort to heal the lesions, or could simply be caused by ectopic *piwi-1*+ or *mat*+ cells pushing muscle cell bodies into a smaller area. In addition to normal enteric muscle, the morphology of the ectopic gut branches appeared normal, with *mat*+ cells surrounding a lumen (Fig 4B). The ectopic gut tissue did not form in isolation, but rather was always connected to the normal gut (Fig 4C). We noted that these gut branches were colocalized with gaps in the pigment cell layer, and thus were likely the unpigmented spots observed in the live animals (S8 Fig). Indeed, feeding the planarians with food dyed green resulted in these white spots taking on a green color, indicating that the ectopic gut branches were connected to the gut at large (S8 Fig).

In addition to the disruption in pigment cell localization, we observed that the regular pattern of epidermal nuclei was altered in regions above a severe muscle lesion, likely indicating the presence of an ectopic gut branch (Fig 4D). Staining with Concanavalin A to mark junctions between epidermal cells revealed that regions of the epidermis with increased DAPI staining did not have regular junctions (Fig 4D). These results suggest that ectopic tissue, most likely gut, invaded into and disrupted the epidermis itself.

## Ectopic stem cells cause lesions in the body wall muscle

The spatial correlation of muscle lesions, ectopic gut branches, and ectopic neoblast clusters suggested a causative relationship between these phenotypes. To examine this possibility, we performed a time course during *megf6* or *hemicentin* knockdown phenotype progression to determine what phenotype appeared first. After 5 feeds of RNAi food and 3 more days of homeostasis (denoted as 5fd3; early-stage phenotype), we observed individual ectopic neoblasts appearing in the plane of the body wall muscle (Fig 5A). This occurred before the

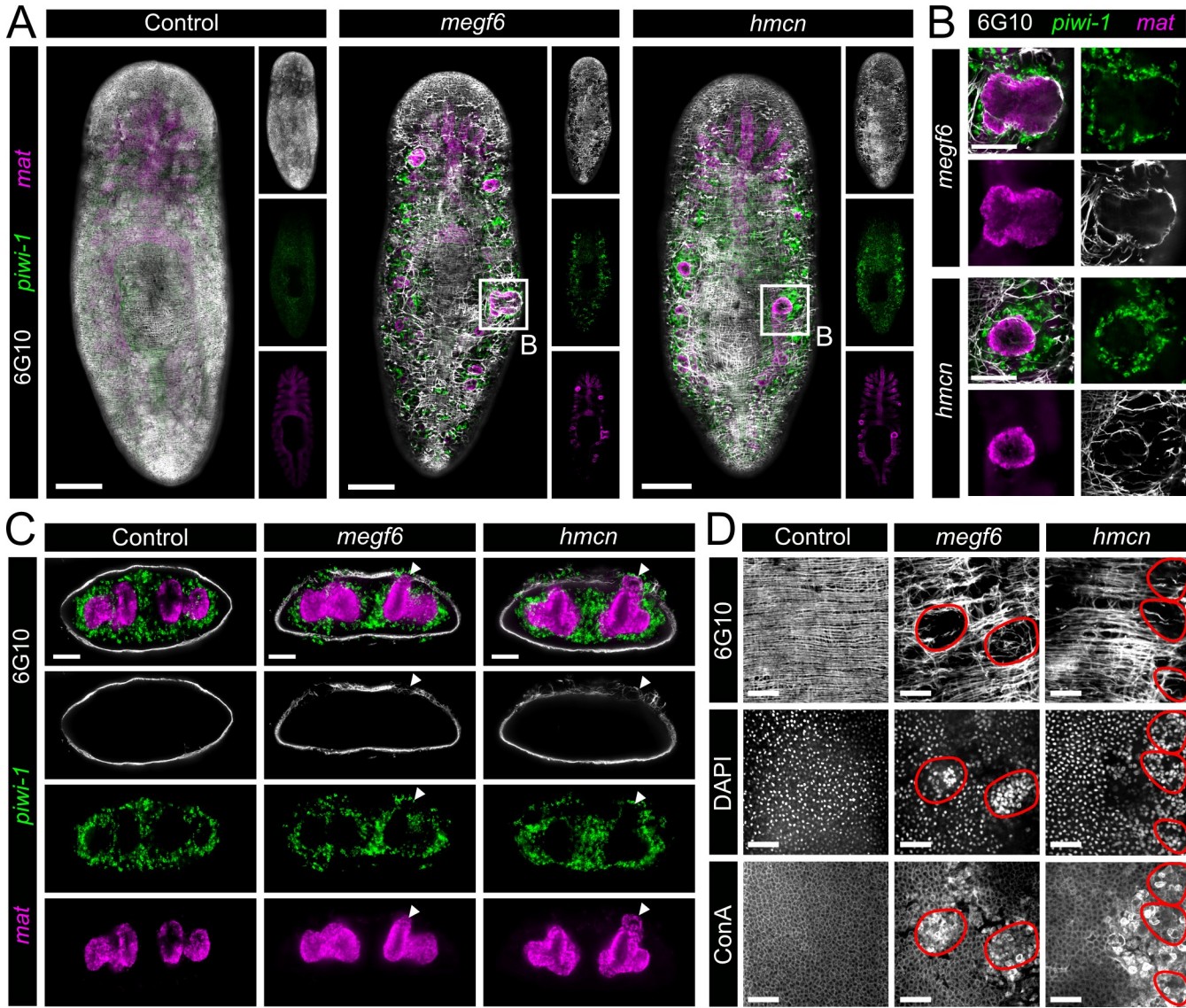

**Fig 4. Muscle lesions and ectopic gut branches associated with ectopic neoblasts. A)** Double FISH for *piwi-1* and *mat* combined with 6G10 antibody to mark muscle fibers (n ≥ 10). Single confocal planes at the dorsal sub-epidermis of whole control, *megf6*, or *hemicentin* knockdown planarians show the dorsal layers of body wall muscle. White boxes denote the magnified regions in B). **B)** High magnification of ectopic *mat*+ cells from the *megf6* or *hemicentin* knockdown animals in A. *piwi-1* and 6G10 stains are also shown. **C)** Transverse cross-sections of control, *megf6*, or *hemicentin* knockdown worms stained with double FISH for *piwi-1* and *mat* and 6G10 antibody. White arrowheads show ectopic *mat*+ structures (n ≥ 3). **D)** 6G10, DAPI, and Concanavalin A (ConA) staining in the same region of control and knockdown planarians (n ≥ 5). DAPI and ConA tiles are the same confocal plane at the epidermis, while 6G10 tiles are extended focus images from the epidermis to the muscle layer directly below. Red circles denote the regions with severe muscle lesions. Scale bars: 250 μm in A; 100 μm in B, C; 50 μm in D.

development of obvious muscle lesions, although the muscle fibers appeared disorganized at this timepoint. By the time clusters of ectopic *piwi-1*+ cells were visible, these clusters were accompanied by clear muscle lesions (7fd3; middle-stage phenotype). Finally, at the late-stage timepoint (9fd3; the timepoint used for all previous experiments), we observed gut branches protruding to the dorsal side surrounded by *piwi-1*+ cells (Fig 5A).

The above results indicated that the gut branches were the last of the three phenotypes to develop, and were therefore unlikely to be causative. In line with this, cross-sections of

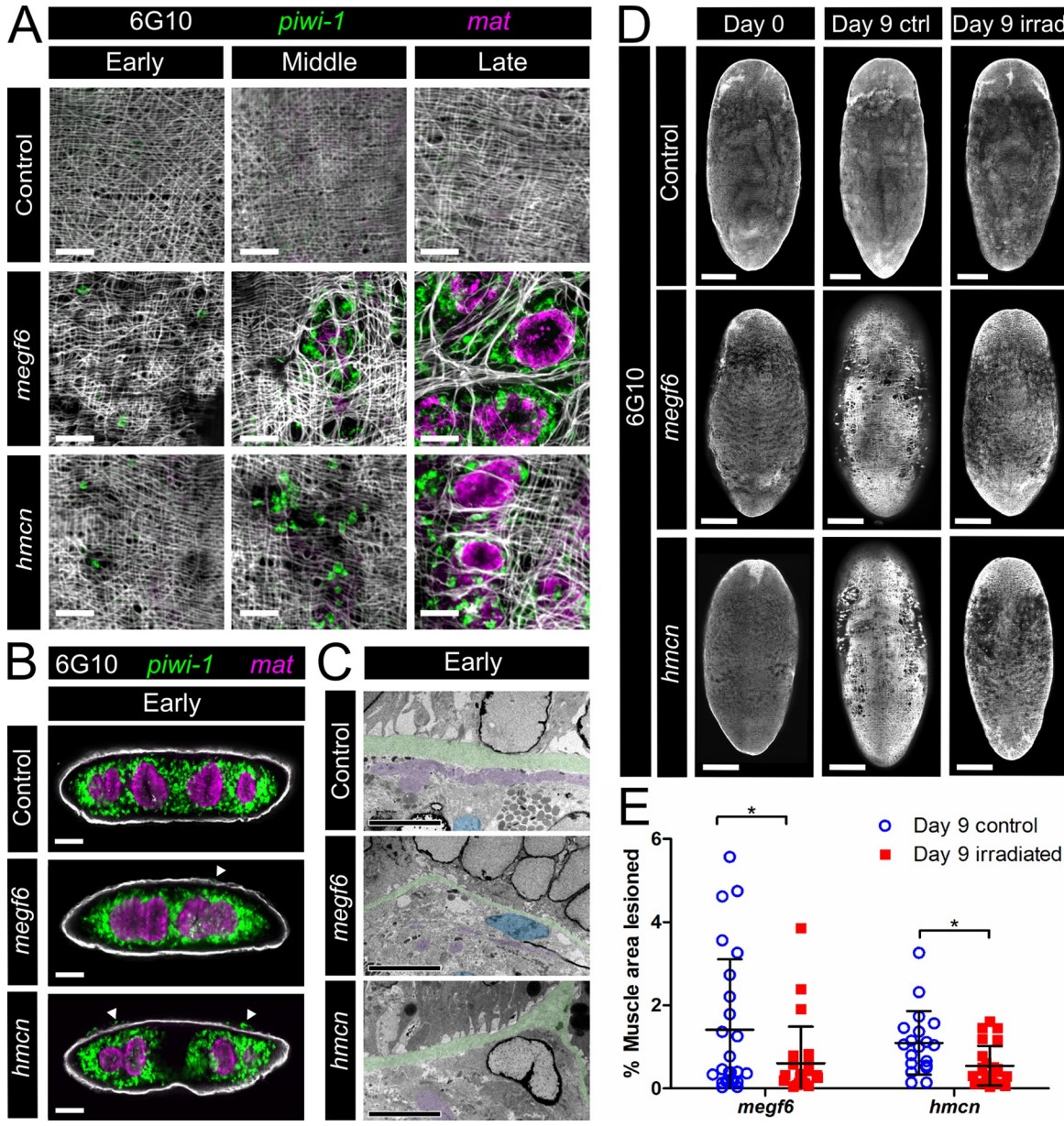

**Fig 5. Development of the *megf6* and *hemicentin* knockdown phenotypes. A)** Timepoints during the progression of the *megf6* and *hemicentin* knockdown phenotypes, shown by double FISH for *piwi-1* and *mat* and 6G10 staining (n ≥ 8). All worms were fixed 3 days after the last RNAi feeding: 5 feeds for the early phenotype, 7 feeds for the middle, and 9 feeds for the late. **B)** Transverse cross-sections of planarians from the early (5fd3) timepoint, stained with *mat*, *piwi-1*, and 6G10 (n = 3). White arrowheads denote *piwi-1* cells above the muscle layer. **C)** Transmission electron microscopy images of planarians from the early timepoint (n = 3). The basement membrane is coloured green, subepidermal nuclei are blue, and muscle fibers are purple. **D)** Confocal extended focus images showing the body wall muscle of control, *megf6*, and *hemicentin* knockdown planarians. Day 0 animals were fixed 3 days after the 4th RNAi feed. At this timepoint, half of the remaining worms were given a lethal dose of irradiation. 9 days later, both irradiated and non-irradiated control worms were fixed and all worms were stained with 6G10 antibody. **E)** Quantification of lesioned area in worms from D), shown as percentage of total muscle area (n ≥ 19). Lesions were quantified relative to controls. Scale bars: 100 μm in A, B; 5 μm in C; 250 μm in D. Error bars are standard deviations. $^*p < 0.05$ (Welch's *t*-test).

planarians at 5fd3 showed normal gut morphology, even directly underneath ectopic neoblasts (Fig 5B). At this timepoint, TEM images showed thinning of the subepidermal basement membrane, indicating that the ECM was disrupted at this early stage (Fig 5C). We

hypothesized that these ECM defects were associated with either the migration of ectopic neoblasts or the formation of muscle lesions, which then resulted in the appearance of the other phenotype. However, it was unclear which of these two phenotypes was causing the other. Previous work has demonstrated that the formation of epidermal wrinkles and appearance of ectopic cells in *hemicentin* knockdown animals occurs independently of stem cells [26]. Therefore, we asked whether muscle lesions could form in the absence of stem cells as well.

To determine whether ectopic neoblasts caused muscle lesioning, we performed lethal irradiation to remove all neoblasts in *megf6* or *hemicentin* knockdown worms at the early-stage phenotype, at the start of the development of muscle lesions. We visualized muscle fibers by immunostaining 9 days post-irradiation and found that the percentage of muscle lesions dropped twofold in irradiated worms compared to non-irradiated controls (Fig 5D and 5E). At this timepoint, gut morphology was normal in all animals, with no ectopic gut branches observed (S9 Fig). This result demonstrated that the presence of neoblasts enhanced the development of muscle lesions in both *megf6* knockdown and in *hemicentin* knockdown animals. Taken together with the observation that muscle lesions do not form in the region above the pharynx or at the tip of the head (regions with few neoblasts), these results support the hypothesis that the ectopic neoblasts drive the development of the lesions.

## Basement membrane integrity is dispensable for regeneration

To assess regeneration in the knockdown planarians, we performed sagittal amputations to directly compare new and old tissue at the exact same axial region of the animal. Interestingly, despite the numerous tissue defects observed in both *megf6* and *hemicentin* knockdown animals, these worms were able to regenerate relatively normally, with muscle, gut, eyes, pharynx, and brain forming as normal (Fig 6A; S10 Fig). Surprisingly, we noted that the newly regenerated tissue appeared free of the epidermal wrinkles, muscle lesions, and ectopic neoblasts observed in the old tissue of the same animal (Fig 6A; S10 Fig). To look for the presence of ectopic cells in the regenerating worms, we stained regenerating sagittal fragments with 6G10 and markers of the pigment cell lineage, then performed transverse cross-sections to visualize the tissue (Fig 6B). After one week of regeneration, the new tissue was clearly defined in both knockdown and control animals by the presence of $PBGD-1^+$ pigment cell progenitors, while the old tissue contained mostly $lysoLP-1^+$ mature pigment cells (Fig 6B) [33]. Intriguingly, the new tissue of *megf6* or *hemicentin* knockdown animals had a smooth layer of body wall muscle, while the muscle layer of the old tissue was clearly disorganized compared to controls. In addition, while mature and immature pigment cells were observed on both sides of the muscle layer in the old tissue, no ectopic pigment cells were observed in the new tissue at one week post-amputation (wpa; Fig 6B). By 2 wpa, $PBGD-1^+$ pigment cells could be seen in line with and beyond the muscle fibers, indicating muscle degeneration and ectopic migration as the homeostatic phenotype re-emerged (Fig 6B).

To determine whether the basement membrane was able to regenerate normally in *megf6* or *hemicentin* RNAi, we performed TEM on new and old tissue of longitudinally regenerating worms (Fig 6C and 6D). We found that the basement membrane of the epidermis was significantly thinner in the new tissue of both *megf6* and *hemicentin* RNAi animals in comparison to control animals, indicating that these genes were required for the proper regeneration as well as maintenance of the basement membrane (Fig 6D). Although the basement membrane was disorganized in the newly regenerated tissue of knockdown animals, muscle fibers were observed immediately adjacent to the basement membrane; in the old tissue of the same animals, the muscle fibers were located several cell diameters from the epidermis (Fig 6C). These results corroborate our observations from the WISH staining, and indicate that both *megf6*

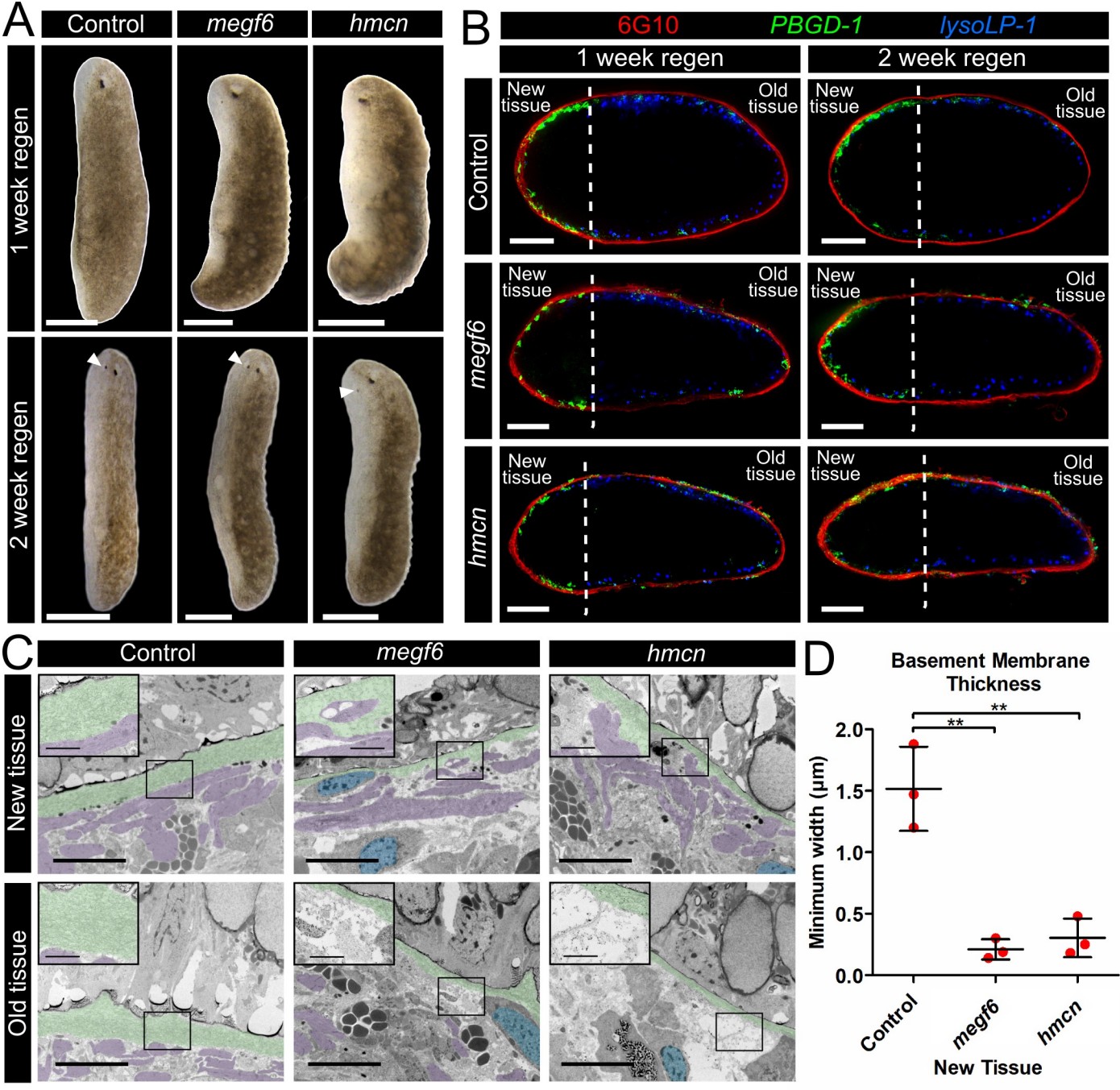

**Fig 6. Tissue defects do not appear in newly regenerated tissue. A)** Live images of regenerating control, *megf6*, or *hemicentin* knockdown planarians 1 or 2 weeks after a sagittal amputation (n ≥ 4). White arrows mark the regenerating eyespots at 2 weeks post amputation. **B)** Cross-sections of worms amputated as shown in A). Worms were stained with double FISH for *PBGD-1* and *LysoLP-1*, and immunostained with 6G10 (n ≥ 6). Amputation planes are marked with dotted lines. **C)** Transmission electron microscopy of the subepidermal region of worms amputated as shown in B) and regenerated for 1 week. The basement membrane is coloured green, subepidermal nuclei are blue, and muscle fibers are purple. Black boxes denote the magnified region, shown in the top left corner of each panel. **D)** Quantification of basement membrane width as visualized in D), measured at the thinnest point in the new tissue of control and knockdown planarians (n = 3). Scale bars: 250 μm in A; 100 μm in C; 5 μm in main panels and 1 μm in insets in D. Error bars are standard deviations. **p < 0.01 (Welch's *t*-test).

and *hemicentin* knockdown animals are able to regenerate new tissue without ectopic cells even in the absence of a normal basement membrane.

# Discussion

## *megf6* and *hemicentin* act throughout the entire organism

The phenotypes produced by *megf6* or *hemicentin* knockdown are extremely diverse, and affect multiple tissues throughout the entire body of the planarian. This is a surprising result, as the expression of *megf6* is restricted primarily to the pharynx, while *hemicentin* is expressed in muscle cells. How can two genes with such different expression patterns produce the same phenotypes upon knockdown? Furthermore, how can these genes affect cell types located far from their source, such as stem cells? Without antibodies to visualize the localization of the proteins, these questions cannot be definitively answered. However, a potential model can be generated with the data available and by extrapolating from other organisms (Fig 7). Disorganization of the epidermal basement membrane in *megf6* or *hemicentin* knockdown suggests a role for these proteins at this location. *Hemicentin*, expressed by muscle cells adjacent to the epidermis, is likely secreted into the basal lamina, and forms a component of the epidermal basement membrane. In other organisms, hemicentin mediates the attachment of adjacent

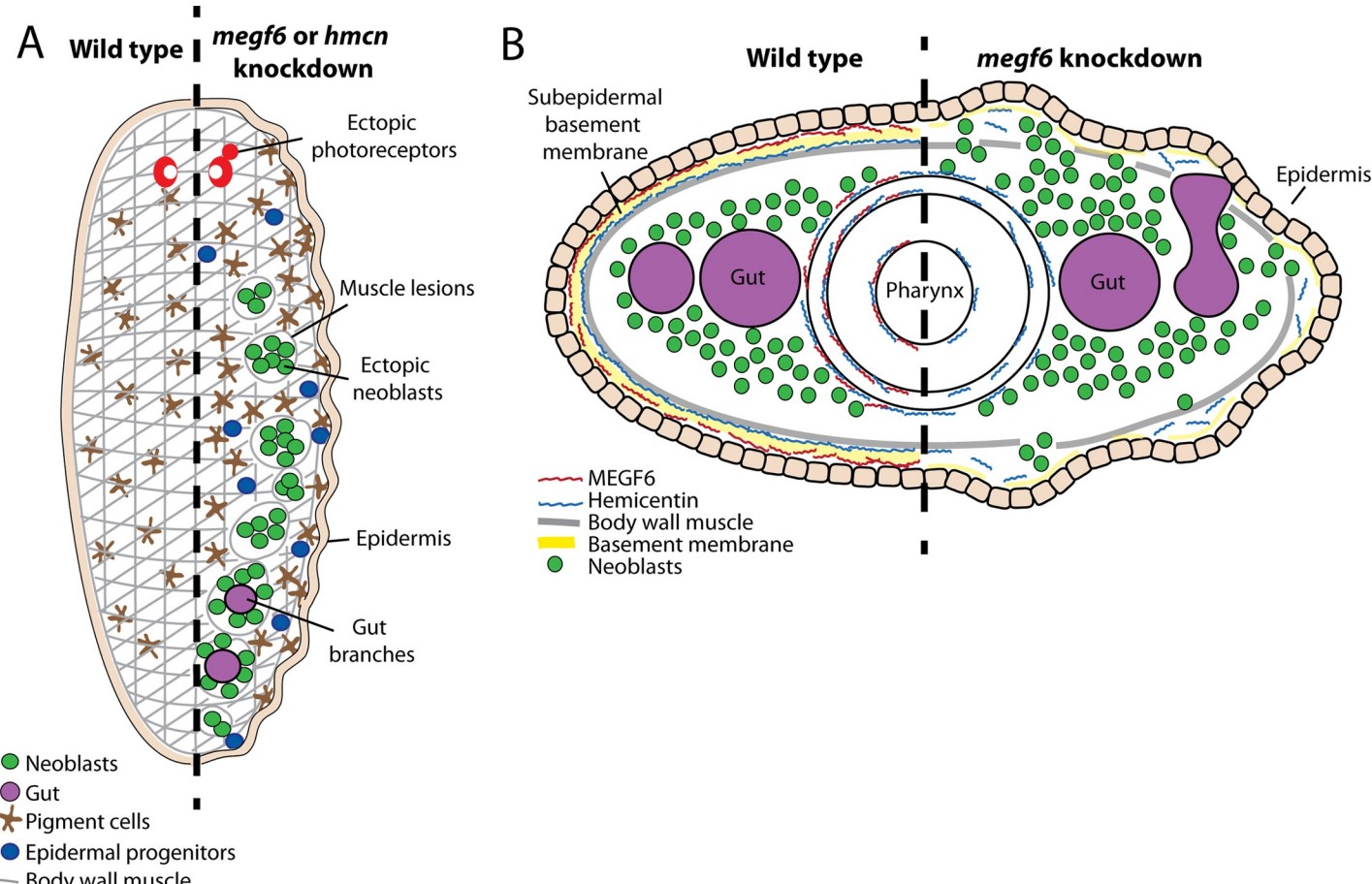

**Fig 7. Model of the localization and knockdown phenotypes of *megf6* and *hemicentin*. A)** Summary of the phenotypes induced by knockdown of *megf6* or *hemicentin*, which include wrinkled epidermis, muscle lesions associated with ectopic neoblasts and gut branches, and mis-localized cell types such as pigment cells, photoreceptors, and epidermal progenitors. **B)** Model of a transverse cross-section showing the proposed locations of MEGF6 and hemicentin proteins, hypothesized based on mRNA expression and knockdown phenotypes. The left half models a wild-type animal, while the right half shows the effect of *megf6* knockdown. Both *megf6* and *hemicentin* are required for proper organization of the subepidermal basement membrane: in the absence of MEGF6 protein, attachment of the basement membrane is impaired, and cell types localize ectopically to the subepidermal space. These cells include neoblasts and gut cells, which normally do not contact the subepidermal basement membrane. The same effects are observed in the absence of *hemicentin*.

basement membranes, as well as linkage of cells to the basement membrane [44,45]. The wrinkled epidermis and ectopic cell types found in *hemicentin* RNAi in planarians suggest that this role is conserved, and *hemicentin* is responsible for the attachment of the muscle layer to the basement membrane of the epidermis (Fig 7). Although *megf6* is primarily expressed in the pharynx, it is also expressed at low levels in cells of the epidermal lineage, suggesting that this molecule may play a similar role in maintaining attachment of the basement membrane.

Impaired attachment of the basement membrane may explain the presence of ectopic pigment cells, which normally reside in the subepidermal space and move through the muscle layer in *megf6* or *hemicentin* knockdown animals. However, the stem cells within their normal compartment do not contact the subepidermal basement membrane, which makes the presence of ectopic stem cells more difficult to explain. This phenotype suggests that *megf6* and *hemicentin* may affect the composition of the ECM throughout the body of the planarian, not just near the cells in which they are expressed. In *C. elegans*, hemicentins have been shown to form long fibers stretching from the body wall muscle to the pharynx, similar to the elastic fibers found in vertebrates [44]. We hypothesize that planarian *hemicentin* might form similar fibers, acting as a scaffold for ECM within the mesenchyme of the worm. These fibers may define the tissue boundaries within the mesenchyme, restricting the stem cells to within their niche, and potentially directing the migration of differentiating cells. Whether *megf6* may also contribute to these fibers is unclear, as this gene has not been previously reported to act as an ECM component. However, planarian *megf6* does bear some structural similarity to the fibulin family of proteins: both contain a series of EGF-like domains [46]. If the MEGF6 protein in planarians acts in a similar way to fibulins in *C. elegans*, it may regulate the assembly and localization of hemicentin fibers, which could explain why *megf6* knockdown phenocopies *hemicentin* knockdown [45,47]. Another possibility is that the EGF-like domains of MEGF6 and hemicentin allow them to directly bind EGFR and mediate growth factor signaling, as has been demonstrated for EGF-containing ECM proteins in other systems [23,48].

## Evolution of the MEGF gene family

Our phylogenetic analysis of the MEGF gene family has demonstrated that both MEGF10/11 and MEGF6 are conserved and ancient genes in metazoans, but MEGF10/11 was lost in Platyhelminthes while MEGF6 was lost in insects. These two separate, clade-specific gene loss events raise the question of how planarians and insects have compensated for the loss of MEGF10 or MEGF6, respectively, particularly given that planarian *megf6* does not seem to recapitulate any of the known roles of MEGF10 in other organisms. In total, the MEGF family of genes has an interesting and previously unappreciated evolutionary history, particularly in highly used laboratory models. Curiously, this has led to the focus on ced-1/Draper/MEGF10/11 homologs, and MEGF6 genes have gone unstudied, with almost no functional data (even in *C. elegans*). Little is known about the role of MEGF6 in any system, though it has been shown to promote the metastasis of colorectal cancer by inducing EMT [49]. Given the phenotypes observed with knockdown of planarian *megf6*, we hypothesize that MEGF6 may have a role in regulating EMT migration through an ECM-based mechanism as described here for planarians.

## Stem cell division and differentiation are regulated by the normal niche

The stem cell niche is generally considered to be the environment capable of maintaining stemness, without which stem cells would die or terminally differentiate [4]. The behavior of planarian stem cells we observed in *megf6* or *hemicentin* knockdown challenges this paradigm. Not only were ectopic stem cells outside of the normal compartment capable of proliferating,

but a higher proportion expressed H3P in comparison to their counterparts within the normal neoblast compartment. These results suggest that in planarians, the hemicentin/MEGF6/basal lamina extent of the niche is restrictive to stem cell growth rather than permissive. If this is the case, signaling from the niche may be an important part of the planarian's impressive resistance to neoplastic-like states. However, it should be noted that other factors, including altered wounding signals or mechanical stress, could contribute to the changes in proliferation we observe, and therefore further characterization of the niche is required to test this hypothesis. Interestingly, we also noted that a significantly smaller proportion of ectopic stem cells expressed *tgs-1*, a putative marker for pluripotency, compared to their counterparts within the normal compartment [15]. This may be due to their distance from the normal intestine, as *tgs-1* is expressed primarily in neoblasts closely associated with the gut [15]. These data suggest that signals in the proximity of the intestine may suppress proliferation, but also act to maintain pluripotency. Future work identifying these signals and their sources within the niche will be needed before this can be stated conclusively.

In the absence of *megf6* or *hemicentin*, ectopic stem cells not only proliferate outside of their compartment but also remodel their environment. These stem cell colonies accelerate the lesioning of the body wall muscle, resulting in large holes in the network of muscle fibers (Fig 5). This may be occurring indirectly through migration of progeny cells from the ectopic clusters, directly due to mechanical stress caused by expanding cell populations, or the stem cells may actively secrete factors such as matrix metalloproteases, similar to how metastasizing cancer cells degrade the ECM [9]. Ectopic stem cells may also cause the ectopic gut branches seen in *megf6* or *hemicentin* knockdown animals, as these gut branches are always found in close association with the stem cells. The planarian gut is responsible for providing nutrients to the tissue of the worm; therefore, formation of ectopic gut branches might parallel angiogenesis, another process induced by cancer cells [50].

## Regenerating tissues organize normally, initially

Despite the severe defects observed in homeostatic tissues of *megf6* or *hemicentin* knockdown animals, these worms are able to regenerate new tissues that appear largely normal. Although the basement membrane in the new tissue is still disorganized, ectopic pigment cells do not appear for at least a week following amputation. Ectopic stem cells, and therefore muscle lesions, are also absent from the new tissue for at least two weeks post amputation. Therefore, although the ECM still appears to be disorganized by *megf6* or *hemicentin* knockdown, this disorganization does not affect the proper regeneration of new tissue.

One explanation for this is that *megf6* and *hemicentin* are responsible for maintaining the integrity of the tissue against the mechanical stresses caused by homeostatic cell turnover, which could explain why ectopic pigment cells begin to reappear by two weeks post-amputation. Another possible explanation for the proper patterning of new tissues in *megf6* or *hemicentin* knockdown worms is that specific changes to the ECM occur following amputation to promote regeneration, resulting in a "regenerative ECM" that does not require the same type of organization as mature ECM. The concept of a regenerative ECM has been described in zebrafish spinal cord regeneration, which requires the deposition of pro-regenerative Collagen XII by fibroblasts [12]. It is possible that amputation triggers the expression of regeneration-specific ECM proteins in planarians, and that these proteins can define tissue boundaries in the absence of *megf6* or *hemicentin*. Understanding how the planarian ECM acts to promote regeneration and suppress excess cell division during homeostasis may help to explain the incredible regenerative and cancer-resistant abilities of the planarian.

## Materials and methods

### Phylogenetic analysis

The software Geneious (www.geneious.com) was used to create a multiple alignment using Clustal Omega plugin. From this, a Bayesian phylogeny was made using the MrBayes plugin and the following settings: unconstrained branch length, shape parameter exponential of 10, 1.1Million chain length, 4 heated chains, 0.2 heated chain temp., WAG substitution model, gamma rate variation model, 10% burnin length with subsampling frequency of 200, with no outgroup. All sequences plus some additional MEGF predicted sequences in Supplementary File 1.

### Animal husbandry and exposure to γ-Irradiation

Asexual *Schmidtea mediterranea* strain CIW4 were reared as previously described [51]. For irradiation experiments, planarians were exposed to 60 Gray (Gy) of γ-irradiation from a $^{137}$Cs source [19].

### RNAi and gene cloning

RNAi experiments were performed using previously described expression constructs and HT115 bacteria [52]. Bacteria were grown to an O.D.600 of 0.8 and induced with 1 mM IPTG for 2 hours. Bacteria were pelleted, mixed with liver paste at a ratio of 333 μl of liver to 100 ml of original culture volume, and frozen as aliquots. The negative control, "Control", was the *gfp* sequence as previously described [53]. RNAi worms were fed 3 times (once every 3 days), then cut into 3 fragments 3 days after the last feed. The worms were allowed to regenerate for 7 days, and then the feeding schedule recommenced up to 9 feeds. Worms were fixed 6 days after the last feeding. This protocol was used for all RNAi worms unless otherwise indicated. All animals were size-matched between experimental and control worms. Sequence homology was determined using PlanMine [54]. Transcript and gene sequences for *megf6* (dd_Smed_v6_5630_0_1) and *hemicentin* (dd_Smed_v6_1161_0_1) can be found at http://planmine.mpi-cbg.de/planmine/begin.do.

### Immunolabeling and in situ hybridizations (ISH)

Whole-mount ISH (WISH) and double fluorescent ISH (dFISH) were performed as previously described [55–57]. Briefly, 5% N-acetylcysteine in phosphate-buffered saline (PBS) was used to kill the worms and remove mucus, followed by fixation in 4% formaldehyde in PBST (0.5% Triton-X) for 20 min. Worms were then rinsed with PBST, further permeabilized with Reduction solution (50mM DTT, 1% NP-40, 0.5% SDS, in PBS) for 5 min, and dehydrated with methanol. Worms were bleached with 6% hydrogen peroxide (in methanol) overnight, and rehydrated with PBST. For ISH, worms were pre-hybridized for 2 h, and then hybridized with probe overnight at 56°C. For triple FISH experiments, probes were made with digoxygenin (DIG), fluorescein (FITC), or dinitrophenyl (DNP) labelling mix. Blocking solution (5% horse serum, 0.5% Roche Western Blocking Reagent, in MABT) was used for blocking and antibody incubation. Probes were detected with anti-DIG-AP (1:4000), anti-DIG-POD (1:500), anti-FITC-POD (1:300), or anti-DNP-AP (1:2000) antibodies. Colorimetric stains were developed using 4-nitro blue tetrazolium chloride (NBT, Roche 11383213001) with 5-bromo-4-chloro-3-indolyl-phosphate (BCIP, Roche 11383221001). FISH stains were developed with either tyramide amplification, or Fast Blue B Salt (Sigma D9805) with naphthol AS-MX phosphate (Sigma 855). Tyramide amplifications were performed in 0.1M borate buffer, pH 8.5 [58]. For FISH in combination with immunostaining, peroxidase inactivation was performed with

100mM NaN$_3$ (in PBST) for 45 min, followed by extensive PBST washes. 6G10 antibody (Developmental Studies Hybridoma Bank product 6G10-2C7, deposited to the DSHB by Zayas, R.M.) was used at 1:1000, and rabbit anti-H3P (EMD Millipore 05-817R-I) was used at 1:1000. Secondary horseradish peroxidase (HRP)-conjugated anti-mouse or anti-rabbit antibodies (Jackson Immunoresearch 115-036-006 and 111-035-144) was used at 1:500, with subsequent tyramide amplification. For H3P immunostaining without *in situ* and Concanavalin A, animals were killed for 10 min in N-acetylcysteine and treated for 2 hours with Carnoy's fixative (6:3:1 ethanol:chloroform:acetic acid). Worms were then bleached, blocked and immunostained as above. FITC-conjugated Concanavalin A (Vector Laboratories, VECTFL1001) was used at a concentration of 5 μg/mL, incubated overnight in blocking buffer.

## TUNEL and BrdU

TUNEL staining was performed as previously described [59] with the Terminal Deoxynucleotidyl Transferase enzyme (Thermo, EP0162). Animals were killed in NAC and incubated in reduction solution as described above, then bleached overnight in 6% hydrogen peroxide (in PBST). Detection was performed as for ISH experiments. BrdU was dissolved in 50% DMSO to a concentration of 50 mg/mL, then mixed with 4 volumes of beef liver and fed to worms raised in 6 g/L Instant Ocean. Staining for BrdU was performed as previously described [60].

## Imaging and quantifications

Live worms, colorimetric WISH stains, and whole-worm H3P stains were imaged on a Leica M165 fluorescent dissecting microscope. dFISH stains were imaged on a Leica DMIRE2 inverted fluorescence microscope with a Hamamatsu Back-Thinned EM-CCD camera and spinning disc confocal scan head. H3P cell counts were quantified using freely available ImageJ software (http://rsb.info.nih.gov/ij/) with the cell counter function, and muscle lesions were labelled by hand, using control RNAi animals as a reference so as to only label muscle gaps larger than those found in controls, and quantified using the ImageJ area measurement function. *Piwi-1*[+] and H3P[+] cell counts were performed in cross-section using Imaris (Bitplane, South Windsor, CT, USA). Cross-sections were taken from the same regions in control and knockdown animals (examples shown in Fig 3A), and "normal" and "ectopic" regions were determined manually using DAPI staining to mark the boundary (as shown in Fig 3B). *Collagen*[+] cells, TUNEL[+] cells, and neoblast subclass markers were also quantified using Imaris. Significance was determined by a two-tailed unequal variance student's *t*-test. All images were post-processed in a similar manner using Adobe Photoshop. Heatmaps were made in R studio using data downloaded from https://compgen.bio.ub.edu/PlanNET/planexp [61].

## Transmission electron microscopy

Animals were dropped into a solution of 2% glutaraldehyde in 0.1M sodium cacodylate, pH 7.3 (EM buffer) and fixed for 2 hours. Samples were incubated in 0.2M sucrose in EM buffer for 10 minutes, then 1% osmium tetroxide in EM buffer for 90 minutes, then sucrose solution again for 10 minutes. Samples were dehydrated in a graded ethanol series followed by propylene oxide and embedded in Quetol-Spurr resin. Transverse sections were cut using a Leica Ultracut ultramicrotome to a thickness of 80 nm, then stained with 2% uranyl acetate and lead citrate. Sections were examined using a FEI Tecnai 20 at 120 kV and imaged using an Orion CCD camera. Sample preparation and imaging were done at the Nanoscale Imaging Facility at the Hospital for Sick Children. False colouring was added manually in Adobe Photoshop.

## Fluorescence-activated cell sorting

Fluorescence-activated cell sorting (FACS) for quantification of the X1 gate was performed as previously described [62]. Briefly, 30 animals per sample were rinsed with cold CMF + 1% BSA (CFMB). Animals were then thoroughly dounced with a sterile plastic pestle and strained through a 40 μm strainer in 3 ml of CMFB. Cells were stained with Hoechst 342 (25 μg/ml) for 10 minutes, then centrifuged (300g for 5 min, low brake) and resuspended in 500 μl of cold CMFB. Cells were sorted on a Beckman-Coulter MoFlo XDP sorter.

## Supporting information

**S1 Fig. Expression of *megf6* and *hemicentin* in distinct tissue types. A)** Heatmap showing the expression of *megf6* and *hemicentin* in different tissues from single-cell RNA sequencing (data from Fincher et al. 2018 and downloaded from https://compgen.bio.ub.edu/PlanNET/planexp). **B)** Double fluorescent *in situ* hybridization for *megf6* or *hemicentin* with markers of the epidermis (*vim-1*), intestine (*mat*), and muscle (*collagen*) in wild-type planarians. White boxes denote the magnified region shown on the right, and white arrows show cells with expression of both genes. **C)** Colorimetric whole mount *in situ* hybridization (WISH) stains for *megf6* and *hemicentin* in control, *megf6*, or *hemicentin* RNAi animals. Scale bars: 50 μm in B; 250 μm in C.
(TIF)

**S2 Fig. A Bayesian phylogeny of MEGF family of proteins.** Names in blue are Deuterostomes, names in red are Ecdysozoans, names in magenta are Lophotrochozoans, and the flatworms are in green. All important node support values are listed as % support next to the relevant node. All MEGF10/11/PEAR/Draper/ced-1 proteins cluster together with 100% support (blue shaded box). A full list of protein sequences and accession numbers can be found in Supplemental File 1. Species names: Lana = *Lingula anatine*[brachiopod]; Pcau = *Priapulus caudatus*[priapulid]; (flatworms) Smed = *Schmidtea mediterranea*; Mcro = *Maritigrella crozieri*[polyclad]; Mlig = *Macrostomum lignano*[basal flatworm]; (Ecdysozoa) Cele = *Caenorabditis elegans*; Dmel = *Drosophila melanogaster*; Tcas = *Tribolium castaneum*[flour beetle]; Pvan = *Penaeus vannamei*[shrimp]; Tscu = *Centuroides sculpturatus*[scorpion]; (Lophotrochozoa) Pdum = *Platynereis dumerilii*[polychaete]; Myes = *Mizuhopecten yessoensis*[mollusk]; Ctel = *Capitella teleta*[polychaete]; (Dueterostomes) Apla = *Acanthaster planci*[starfish]; Spur = *Strongylocentrotus purpuratus*[sea urchin]; Cmil = *Callorhinchus milii*[shark]; Blfo = *Branchiostoma floridae*[lancelet]; Xtro = *Xenopus tropicalis*; Hsap = *Homo sapiens*; Drer = *Danio rerio*.
(TIF)

**S3 Fig. Comparison of the knockdown phenotypes of *megf6* with the Draper-pathway genes *ABCA-1*, *ELMO*, *gulp*, and *shark*.** Single confocal planes showing whole mount fluorescent *in situ* hybridization (FISH) for *PBGD-1*, FISH for *piwi-1*, and immunostaining for muscle fibers with 6G10 antibody (n ≥ 8). Draper pathway knockdown animals are compared with controls (left) as well as *megf6* knockdown planarians (right). Scale bars are 250 μm.
(TIF)

**S4 Fig. Whole animal quantification of stem cell number and proliferation. A)** Quantification of *piwi-1* cells in transverse cross-sections shown in Fig 3A (n ≥ 24). Cross-sections were taken from the same axial regions of control and knockdown animals. Diagram shows the regions quantified (left). **B)** Whole mount staining for phosphorylated-histone H3 on serine 10 (H3P) in control, *megf6*, or *hemicentin* knockdown worms (n ≥ 30). **C)** Quantification of

the number of mitoses, measured from the whole animal as shown in A). **D)** FACS plots of Hoechst-stained cells from control, *megf6*, or *hemicentin* knockdown worms. The X1 gate contains actively cycling neoblasts with >2n DNA content. Proportions of cells in the X1 and X2 gates are shown on each plot. Scale bars are 250 μm. Error bars are standard deviation. $^*p < 0.05$, $^{***}p < 0.001$ (Welch's *t*-test).
(TIF)

**S5 Fig. Quantification of neoblast subclass markers.** Percentage of *piwi-1*+ cells co-expressing marker genes of the zeta (*zfp-1*), gamma (*hnf4*), or putative pluripotent (*tgs-1*) neoblast subclasses in control and knockdown planarians (n ≥ 8). Quantifications are of dFISH images as shown in Fig 3D, with 'normal' and 'ectopic' compartments taken from different confocal planes equivalent to the regions shown in Fig 3B. Error bars are standard deviation. $^*p < 0.05$, $^{**}p < 0.01$, $^{***}p < 0.001$ (Welch's *t*-test).
(TIF)

**S6 Fig. Phenotypes on the ventral side of knockdown animals.** Single confocal planes showing a ventral view of control, *megf6* or *hemicentin* knockdown planarians (n ≥ 10). The animals are stained with double FISH for *piwi-1* and *mat* and immunostained with 6G10 to mark muscle fibers. Scale bars are 250 μm.
(TIF)

**S7 Fig. Muscle cells in knockdown of *megf6* or *hemicentin*. A)** WISH of *collagen-1*, marking the cell bodies of body wall muscle. Red boxes mark the magnified regions shown in the insets. **B)** Quantification of $collagen^+$ cells/mm$^2$ in control, *megf6*, or *hemicentin* knockdown animals (n ≥ 7). Cell densities were measured from 20x tiles as shown in the insets of A. **C)** Whole worm images of TUNEL staining in control and knockdown planarians. **D)** Quantification of $TUNEL^+$ cells from whole worm images shown in C (n ≥ 13). Scale bars are 250 μm, 100 μm for insets. Error bars are standard deviation. $^*p < 0.05$, ns = not significant (Welch's *t*-test).
(TIF)

**S8 Fig. Ectopic gut branches correspond with gaps in pigment. A)** Double FISH for *PBGD-1* and *mat* in *megf6* or *hemicentin* knockdown planarians (n ≥ 6). Single confocal planes at the dorsal side of the animal are shown. **B)** Live image of a *megf6* knockdown planarian after feeding with dyed liver (n = 1). The worm is shown dorsal side up, anterior to the right. Scale bars are 250 μm.
(TIF)

**S9 Fig. Gut morphology in irradiated *megf6* or *hemicentin* RNAi animals.** Single confocal planes showing the intestine, marked by *in situ* hybridization for *mat*, in animals from the experimental timepoints shown in Fig 5D (n ≥ 19). Scale bars are 250 μm.
(TIF)

**S10 Fig. Regeneration of sagittal fragments of control and knockdown planarians.** Optical sections of regenerating fragments 1 or 2 weeks following sagittal amputation (n ≥ 3). Structures shown are muscle fibers (6G10 immunostaining), neoblasts (FISH for *piwi-1*), intestine (FISH for *mat*), brain and pharynx (DAPI). Amputation planes are denoted with dotted red lines. Scale bars are 250 μm.
(TIF)

**S1 File. FASTA text file of the sequences and accession numbers in S2 Fig.**
(TXT)

**S1 Data. Raw count data for specific quantifications in the associated Figure files.**
(XLSX)

## Acknowledgments

We would like to thank Dr. Stephan Q. Schneider for helpful discussions regarding the phylogenetics of the MEGF family.

## Author Contributions

**Conceptualization:** Nicole Lindsay-Mosher, Bret J. Pearson.

**Data curation:** Nicole Lindsay-Mosher, Andy Chan.

**Formal analysis:** Nicole Lindsay-Mosher, Andy Chan.

**Funding acquisition:** Bret J. Pearson.

**Investigation:** Nicole Lindsay-Mosher, Andy Chan.

**Methodology:** Nicole Lindsay-Mosher, Bret J. Pearson.

**Project administration:** Bret J. Pearson.

**Resources:** Bret J. Pearson.

**Supervision:** Bret J. Pearson.

**Validation:** Nicole Lindsay-Mosher, Andy Chan.

**Writing – original draft:** Nicole Lindsay-Mosher, Bret J. Pearson.

**Writing – review & editing:** Nicole Lindsay-Mosher, Bret J. Pearson.

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
