## [Decision Letter · Decision Letter 0]

17 Oct 2019

Dear Dr Pearson,

Thank you very much for submitting your Research Article entitled 'Planarian EGF repeat-containing genes draper and hemicentin are required to restrict the stem cell compartment' to PLOS Genetics. Your manuscript was fully evaluated at the editorial level and by independent peer reviewers. The reviewers appreciated the attention to an important problem and praised the quality of work as well as the overall impact of findings, but raised some substantial concerns about the current manuscript. Based on the reviews, we will not be able to accept this version of the manuscript, but we would be willing to review again a much-revised version. We cannot, of course, promise publication at that time.

If you decide to revise the manuscript for further consideration at PLOS Genetics, please aim to resubmit within the next 60 days, unless it will take extra time to address the concerns of the reviewers, in which case we would appreciate an expected resubmission date by email to plosgenetics@plos.org.

[LINK]

We are sorry that we cannot be more positive about your manuscript at this stage. Please do not hesitate to contact us if you have any concerns or questions.

Yours sincerely,

Christian Petersen, Ph.D.

Guest Editor

PLOS Genetics

Gregory P. Copenhaver

Editor-in-Chief

PLOS Genetics

Reviewer's Responses to Questions

**Comments to the Authors:**

Reviewer #1: This manuscript focuses on the role of extracellular matrix in planarians, and investigates the role of ECM in controlling tissue organization in planarians. Planarian muscle is the source of many, if not all, ECM proteins (as shown in Cote et al, 2019). Here, the authors investigate the role of EGF-containing proteins draper and hemicentin. Hemicentin has been shown previously to constrain distribution of stem cells and their progeny. Knockdown of draper results in a virtually identical phenotype to hemicentin RNAi, where stem cells and intestinal branches are no longer contstrained by the ECM/muscle periphery, and instead leak out beyond the ECM. It is very interesting that the expression of draper is so different from hemicentin, yet their knockdown phenotypes are so similar. The authors document this phenotype with beautiful, clear imaging of several disrupted cell types. The stem cells, which are normally restricted to the interior of the body adjacent to intestines, now appear on the outside of the body and continue to proliferate there. The authors suggest that stem cells initiate rupture through the muscle layer, because stem cells emerge prior to the intestinal branches, and the lesioned area is slightly less in irradiated than in control animals. Overall, this is a very intriguing manuscript with very clear and thorough documentation of the phenotypes. However, some of the conclusions are slightly overstated and would benefit from clarification as suggested below.

Major comments:

It remains unclear what is the primary driver of the phenotype. The radiation experiment suggests that stem cells play some role, but the subtlety of the phenotype suggests that other factors are involved too. It seems like the primary difference between the new and old tissue is that muscle fibers no longer attach to the ECM. One way to characterize this further would be to determine when the intestinal branches emerge out of the newly regenerated tissue, and examine the muscle/ECM proximity by EM prior to onset of that phenotype. If muscle and ECM are still adjacent to each other, then the authors can more conclusively state that the stem cells drive the phenotype.

Stem cells are often located adjacent to the intestine, so as the intestine ruptures through the muscle wall, the stem cells may come along passively, giving the impression that they are leading an invasive event. Is there a way to rule out this possibility?

The conclusion that draper and hemicentin knockdown animals display hyperproliferation is poorly supported. The methods for quantification should be better explained to justify how the “ectopic niche” is distinguished from a “normal niche” – for example, is it restricted to the outer epidermis? Without this information, it is not clear, at least to this reviewer, what the criteria for hyperproliferation were. The authors should clarify this issue, particularly because in the control image (3A), there is proliferation occurring just below the epidermis.

In other organisms, Draper functions in apoptotic cell engulfment. However, it seems to have a different role in planarians – no transmembrane domain, and a phenotype distinct from that of other known engulfment proteins. In this sense, it almost seems disingenuous to refer to this gene as a Draper homolog, and instead the authors may consider referring to it as MEGF6.

The radiation experiment should include intestinal morphology as a control.

The authors should mention if/when the animals die, and how this relates to the number of RNAi feeds administered. Also, population sizes and penetrance need to be indicated.

Reviewer #2: The manuscript by Lindsay-Mosher et al. entitled “Planarian EGF repeat-containing genes draper and hemicentin are required to restrict the stem cell compartment” examines the role of the ECM in planarians stem cell regulation. Knockdown of the aforementioned EGF-containing genes results phenotypes wherein uninjured planarians develop epidermal wrinkling and the formation of unpigmented regions/spots on the dorsal side. Analysis of the cellular processes underlying the observed phenotypes indicate that the ECM components studied in this paper play a role in restricting the localization of stem cells in the parenchyma. Absence of draper and hemicentin leads to an ectopic accumulation of stem cells as well as the ectopic differentiation of intestinal branches. Overall, this is a very nice story founded on beautiful and compelling data, which also supports findings reported in a recent screen by Cote et al. However, the paper is descriptive in its current form, and lacks a clear hypothesis and strong conclusions. In my opinion, additional experiments will be necessary to substantiate some of the conclusions and provide some mechanistic insight into the role of the ECM in neoblast regulation. Furthermore, some of the terminology co-opted in the paper seemed inadequate. My concerns and suggestions for improving the paper are described below.

Major concerns:

The abstract and introduction hint at a goal of uncovering mechanisms restricting cell division and the localization of the planarian neoblasts. But the manuscript focuses on describing the phenotypes rather than analyzing the source of the ectopic neoblasts or the signaling mechanisms leading to the expansion and clustering of the stem cell population. I think the authors are correct in stating that the stem cell niche is not well defined in planarians. Knocking these two ECM components (in contrast to other factors, like beta1-integrin), convincingly affects the stem cell population. It would be much more insightful to explore which stem cell subtypes are affected by the knockdown and whether there are truly more neoblasts (if there is hyperproliferation) or ectopic/mispositioned stem cells. Thus, the authors present interesting ideas regarding understanding how an [undefined] niche restricts localization of the stem cell population, which likely indirectly impacts their regulation (i.e., cell division rates, differentiation signals) but do not explore this problem in depth. The types of experiments that could be performed include in situs for neoblast subtypes, flow cytometry measurements, and TUNEL (which was conspicuously missing from the post-RNAi analysis where lesions are described).

Specific comments/suggestions:

1) The description of draper homology and its expression could be improved. The authors vacillate between ascribing draper expression to pharynx cells and suggesting it is likely in other cell types. The t-SNE plot reveals that draper is highly expressed in other cell types. What are they? The lack of clarity about draper localization becomes problematic for the conclusions and the model. As the authors describe in the Discussion, it doesn’t make much sense that pharynx expression would cause the observed phenotype. Furthermore, I didn’t see a solid phylogenetic analysis for draper, which combined with the evidence described in the paper (e.g., previous literature, lack of similar expression to other draper genes and function, and the domain structure) indicate to me that the gene analyzed in this paper is not really a draper ortholog. From what I gathered in the Discussion, the authors could consider naming this gene Smed-megf6, which according to the description of the structure appears to be a better fit for the planarian homolog. I think the Pearson lab is adept at analyzing gene evolutionary relationships. Such analysis is a glaring omission from this paper.

2) The authors describe the formation of white spots/unpigmented regions on the dorsal surface as lesions. Can the authors visualize epidermal openings? To address this distinction between their conclusions and Cote et al., the authors could stain RNAi knockdown planarians with fluorescent-Concanavalin A to visualize the epidermal surface of the animals in those regions. From the data shown here and in the Cote manuscript, I would refrain from calling the spots lesions.

3) The paragraph discussing epidermal progression after RNAi: I think the authors should be cautious with the interpretation that the “progression” of epidermal differentiation is disrupted. From the data presented I would conclude localization of the cells is disrupted, unless experiments are performed to assess progression (e.g., BrdU pulse-chase labeling experiments).

4) Section on ectopic neoblast hyper-proliferation and Figure 3: this is a key figure in this paper and where I think the authors would consider performing more in-depth analyses. The observations that knocking down “draper” and hemicentin leads to ectopic neoblast localization and that mis-localized neoblasts have proliferative potential are not debatable and are consistent with observations from the Reddien lab. However, the intriguing possibility that ectopic neoblasts divide at a higher rate is not as convincing (Figure 3B and Figure S6). One of the most exciting observations in the manuscript is the formation of piwi-1+ clusters in RNAi knockdown conditions. This suggests that there are more neoblasts in the regions analyzed and I suspect one is more likely to detect dividing cells within those clusters. Unfortunately, it is difficult to assess how this analysis was performed. The authors could show images of the regions analyzed and should normalized the cell counts by counting the number of piwi-1+ cells within the region of interest and the area or volume (unless the regions of interest quantified were the same between control and experimental worms). A related question that I have is whether there are more stem cells in ECM RNAi worms, which could be assessed by performing flow cytometry analysis; if so, this would provide strong support for a hyper-proliferation phenotype. Additional experiments that would augment the impact of this part of the story would be to include labeling for stem cell subtypes following RNAi. For example, does the distribution of tspan-1+ (or another marker like tgs-1) stem cells change and could it explain how the formation of ectopic neoblast “colonies” form? Do they also associate with ectopic gut branches? Finally, I am a little skeptical about the observations presented in Figure S6. If one ignores the statistical tests, there is not a robust difference in cell division rates. I was also wondering why the authors chose to perform a Welch’s t-test for this data, which I believe is most appropriate when samples have unequal samples sizes and/or variances. If they were to perform a standard Student’s t-test, are the differences still significant? In any case, my interpretation in the whole animal analysis is that although significant (as shown), there is no strong support for global changes in proliferation rates if one were to compare these phenotypes to others that clearly lead to altered cell division rates and expansion of the neoblast population, a parameter that could be analyzed by FACS.

5) What is the normal niche? I don’t think this is well-defined in planarians, other than localization of neoblasts near the intestinal branches. I would consider omitting the word niche when describing the phenotype.

6) Although there is evidence for ectopic cell localization (as in tumors) to cause lesions, I am not convinced about the use of this word throughout the paper (as mentioned in Comment #2). What is the evidence the cells are “lesioning” the muscle? Are they simply causing muscle to be disorganized or to become remodeled as a result of ectopic differentiation of other organs, like the intestine? Moreover, Figure S8 shows an increase in the number of collagen+ cells. The authors mention that this could be also due to an increase in muscle cell death. Although I am not aware if the muscle antibody is compatible with the TUNEL assay, the authors could test if there is a global or regional increase in cell death.

Minor points:

1) The author summary contains a few typos.

2) The Methods section is unacceptably superficial. There is no explanation of how cell counts were performed for Figure 3B or for the triple-FISH experiments shown in Figure 2.

Reviewer #3: In this manuscript entitled “Planarian EGF repeat-containing genes draper and hemicentin are required to restrict the stem cell compartment,” Lindsay-Mosher et al. expand our understanding of the diverse functions of ECM components in regulating stem cells. Using the planarian Schmidtea mediterranea as a model, the authors have investigated two genes, draper and hemicentin, which putatively encode ECM proteins required for basement membrane integrity, organization of the sub-epidermal muscle layer, and restriction of localization/growth of stem cells, stem cell progeny, intestine, and eyes in uninjured animals. Both genes encode EGF repeat-containing proteins. Consistent with their phenotypes, hemicentin is expressed in pharynx and subepidermal muscle, while draper is expressed in pharynx and epidermal cells, apparently throughout the animal. hemicentin’s phenotype was previously described in a genome-wide analysis of the planarian matrisome (Cote et al., Nat. Comm., 2019); Lindsay-Mosher and colleagues’ studies largely replicate and validate this previous study, but with additional analysis and insights (e.g., hyperproliferation of mis-localized neoblasts, ectopic growth of gut and eyes were reported in the current manuscript). The studies of draper are novel. Intriguingly, its knockdown phenocopies hemicentin RNAi (including epidermal ruffling, muscle disorganization, and body wall lesions), suggesting draper has a role in maintaining ECM and tissue boundaries, rather than in phagocytosis, its better characterized role in other model organisms. The authors conduct very detailed analyses of the draper and hemicentin phenotypes, and suggest their data indicate that (a) draper and hemicentin restrict migration/growth of multiple tissues, (b) that piwi-1-positive neoblasts “cause” ectopic outgrowths, (c) that draper, hemicentin, and normal ECM are dispensable for at least seven days of regeneration, and (d) that the “normal” niche in planarians inhibits proliferation.

The manuscript is clearly written, experiments are conducted rigorously, and figures are clear and organized. The study will be of interest to the planarian research community, since only a few studies have identified genes that restrict the stem cell compartment, and this model is ideally suited to address this understudied problem. Furthermore, the study should also attract broader interest by those interested in ECM biology and how ECM influences stem cell dynamics and tissue regeneration. However, the authors are encouraged to consider several concerns, both major and minor, that would significantly improve the manuscript if addressed. These issues include the depth of analysis of whether these genes are dispensable for regeneration, the ultimate cause of body wall lesions, interpretation of draper’s roles with respect to phagocytosis vs. functioning as an ECM component, and interpretation of whether the “normal” niche restricts proliferation.

Major concerns:

(1) Are draper and hemicentin really dispensable for regeneration? The authors’ conclusion is based on quite superficial characterization of the RNAi phenotypes in new tissue, especially given that the ECM is not normal in new tissue (Fig. 6D-E). For example, although Fig. 6A shows regeneration of anterior and posterior muscle, it is not clear if epidermis, brain, eyes, or intestine regenerate properly. Similarly, although the experiment in Fig. 6B (sagittal amputation) is elegant, it is not clear if muscle, epidermis, brain, eyes, intestine, or pharynx regenerate properly. The authors only show that ectopic pigment cells arise at 2 weeks in the new tissue (even though close inspection suggests there might be a small number of externally located pigment cells at 1 week). At a minimum, the authors should show images of epidermis (is there evidence of wrinkling?) and eyes from animals in Fig. 6A, and whole mount images of muscle labeling in new vs. old tissue in Fig. 6B, as well as images of epidermis and eyes. It would also be straightforward to assess CNS and pharynx regeneration in sagittally amputated animals with DAPI labeling. While possibly beyond the scope of the manuscript, obviously in situ hybridization would be the best way to assess regeneration of other organs, and/or whether ectopic stem cells and/or progeny emerge earlier.

(2) Do piwi-1-positive neoblasts really cause body wall lesions? The authors attempt to determine the cause of body wall lesions by conducting a detailed time course of the emergence of phenotypes in draper and hemicentin RNAi animals, and by ablating stem cells using irradiation. However, the authors do not seem to consider the possibility that stem cell progeny in the epidermal lineage (e.g., prog-2-positive, agat-3-positive), and not stem cells themselves, are responsible. This seems likely given that prog-2 and agat-3-positive cells (and not piwi-1-positive neoblasts) normally must migrate through sub-epidermal muscles and basement membrane, making them likely candidates to initiate body wall lesions through secretion of MMPs or other processes. In Fig. 2B, many more prog-2-positive and agat-3-positive cells are observed in the epidermis and subepidermis in draper and hemicentin knockdowns, supporting this idea. The authors should test this possibility experimentally, or soften their interpretation and consider this possibility in the results and discussion.

(3) Does dysregulation of phagocytosis contribute to the draper phenotype? The authors suggest that draper’s main function is to regulate ECM, since knockdown of other genes with putative roles in phagocytosis does not have the same phenotype, and since draper does not encode a transmembrane domain. However, this is not a formal demonstration that draper is or is not required for phagocytosis. Isn’t it possible that failure to phagocytose old (senescent?) or ectopically located cells may contribute to ECM degradation and/or muscle disorganization? Have the authors tested whether old or dying cells accumulate in draper RNAi animals? The authors might want to consider this idea more extensively in their Discussion, or preferably, test it directly.

(4) Does the “normal” niche restrict proliferation? The authors report that proliferation (based solely on H3P-S10 labeling) is elevated in the “ectopic” niche (which is only vaguely defined by the authors, Fig. 3B, see below in minor concerns), while proliferation is unaffected in the “normal” niche (again, only vaguely defined). Based on this, the authors suggest (in the abstract, intro, and discussion) that the “normal” niche and/or the basement membrane itself may suppress stem cell proliferation. The authors seem to consider this the only interpretation of their results. Given that no molecule that suppresses or promotes proliferation in the “ectopic” niche has been identified, and that proliferation is unaffected in the “normal” niche despite the fact that some piwi-1-positive cells are located immediately basal to the muscle layer, the authors’ interpretation might be perceived as highly speculative to some readers. While the observations are intriguing, it might be prudent to consider additional explanations in discussion. For example, isn’t it possible that disorganized/ectopic muscle cells secrete pro-proliferative cues inappropriately? Or that ectopic gut branches (or visceral muscle) provide growth-promoting nutrients or other cues (considered superficially on line 385)? Or, if draper does regulate phagocytosis, that elevated populations of dying cells might promote proliferation?

Minor concerns:

(1) Lines 206-207 & Fig 3A-B. The authors should more clearly (and quantitatively, if possible) define “normal” vs. “ectopic” niche regions on the sections in which cells were quantified, preferably with a diagram. Were neoblasts basal to the basement membrane included as ectopic, or not? Experimental precision here is very important to the authors’ conclusions regarding a possible suppressive role for the niche.

(2) Greater detail is required in Fig. 5C for how “% muscle area lesioned” was quantified, since this is a quantitative experiment. Also, values for the control animal should be included.

(3) More detail in methods is required for determining homology of planarian draper to its Drosophila and human homologs. Was this determination made via BLAST? If so, which regions are responsible for homology? What settings were used in SMART to identify different domains? Is it possible to provide a molecular phylogenetic tree?

(4) Why are there increased muscle cells? Is this due to increased differentiation, or reduced cell death? This might warrant brief discussion. Also, are increased muscle cells observed ventrally?

(5) Fig. S7. An image of the ventral muscles in a control animal should be provided.

(6) Lines 154-157, lines 176-186, and/or in Fig. 5B. Did the authors observe wrinkled epidermis in irradiated animals, as in Cotes et al. - reference [26]? Although a minor issue, this might warrant brief discussion, since it would seem that epidermal detachment from the BM occurs independently of ectopic migration of neoblasts and their progeny.

(7) Lines 245-246. The authors mention that the gut branches were the “cause” of the white lesions observed in the animals. Would it be better to say that the lesions “are” gut branches visible immediately below the epidermis, since the authors suggest that neoblast “cause” the lesions (notwithstanding Major concern #2 above)?

(8) Fig. 1 and 6. Would the authors consider including a cross-section schematic of the defects to accompany the TEM images for readers unfamiliar with planarian anatomy?

(9) What is the ultimate fate of draper and hemicentin RNAi animals? Do they die/lyse?

**Have all data underlying the figures and results presented in the manuscript been provided?**

Reviewer #1: None

Reviewer #2: Yes

Reviewer #3: Yes

PLOS authors have the option to publish the peer review history of their article (what does this mean?). If published, this will include your full peer review and any attached files.

Reviewer #1: No

Reviewer #2: No

Reviewer #3: No

---

## [Decision Letter · Decision Letter 1]

16 Jan 2020

Dear Dr Pearson,

We are pleased to inform you that your manuscript entitled "Planarian EGF repeat-containing genes megf6 and hemicentin are required to restrict the stem cell compartment" has been editorially accepted for publication in PLOS Genetics. Congratulations!

Please note Reviewer #3 has a few minor textual comments to consider as you prepare your final draft for production team.  With regard to their first comment, PLOS Genetics doesn't allow any form of "data not shown", so while it would be desirable to indicate the number of genes screened, please take care in your wording.  The editors will not need to reevaluate these minor changes.

Yours sincerely,

Christian Petersen, Ph.D.

Guest Editor

PLOS Genetics

Gregory P. Copenhaver

Editor-in-Chief

PLOS Genetics

Comments from the reviewers (if applicable):

ECM-derived signals are an important area of investigation in understanding the regulation of stem cells and regenerative abilities. The reviewers agree that this revised manuscript now makes a novel and well-supported contribution resolving functions for the factors megf6 and hemicentin in control of the localization and activities of planarian neoblast stem cells. These findings uncover new genetic mechanisms by which ECM signaling constrains the activity of pluripotent adult stem cells, and so they are of significant general interest to the stem cell and regeneration fields.

Reviewer's Responses to Questions

**Comments to the Authors:**

Reviewer #1: With the additional quantifications, schematics, and EM, the authors have addressed my major concerns about this manuscript. The images are beautiful. By toning down the 'engulfment' and 'niche' assumptions that were previously based on unclear homology, the authors have made a more reasonable contribution to the literature.

Reviewer #2: The authors have addressed reviewer concerns and have added extensive new data that have facilitated interpretation of the data and strongly support the conclusions. This version of the manuscript is significantly improved and I believe that is is suitable for publication in PLOS Genetics.

Reviewer #3: The manuscript has been significantly improved and strengthened due to the addition of phylogenetic analysis of the MEGF6/10 protein family, clarification of existing data and methods, provision of new data, and improvements to the discussion.

All concerns raised in the first review have been adequately addressed. There are a few very minor concerns in the revised manuscript that the authors might wish to address:

(1) The authors have moved the mention of their screen of ECM components to the beginning of Results from the Introduction, without providing the number/identity of genes screened or results of the screen unrelated to hemicentin and MEGF6. Would it be appropriate to add a statement that "results to be presented elsewhere" and/or a superficial indication of how many genes were screened/yielded phenotypes?

(2) Line 139 -- The authors say hemicentin "doubled" with collagen+ muscle cells. This is lab-specific description that should be clarified.

(3) Line 514 (and the original MS) -- Carnoy's "fixative" is made with ethanol, while methacarn is made with methanol.

**Have all data underlying the figures and results presented in the manuscript been provided?**

Reviewer #1: Yes

Reviewer #2: Yes

Reviewer #3: Yes

PLOS authors have the option to publish the peer review history of their article (what does this mean?). If published, this will include your full peer review and any attached files.

Reviewer #1: No

Reviewer #2: No

Reviewer #3: No

**Data Deposition**

http://datadryad.org/submit?journalID=pgenetics&manu=PGENETICS-D-19-01456R1

**Press Queries**

---

## [Editor Report · Acceptance letter]

13 Feb 2020

PGENETICS-D-19-01456R1 

Planarian EGF repeat-containing genes megf6 and hemicentin are required to restrict the stem cell compartment 

Dear Dr Pearson, 

We are pleased to inform you that your manuscript entitled "Planarian EGF repeat-containing genes megf6 and hemicentin are required to restrict the stem cell compartment" has been formally accepted for publication in PLOS Genetics! Your manuscript is now with our production department and you will be notified of the publication date in due course.

With kind regards,

Matt Lyles

PLOS Genetics

On behalf of:
